# Low-temperature strain-free encapsulation for perovskite solar cells and modules passing multifaceted accelerated ageing tests

Paolo Mariani[1,8], Miguel Ángel Molina-García[2,8], Jessica Barichello[1], Marilena Isabella Zappia[2], Erica Magliano [1], Luigi Angelo Castriotta [1], Luca Gabatel [2,3], Sanjay Balkrishna Thorat[2], Antonio Esaú Del Rio Castillo[2], Filippo Drago [4], Enrico Leonardi[5], Sara Pescetelli [1], Luigi Vesce[1], Francesco Di Giacomo[1], Fabio Matteocci [1], Antonio Agresti [1], Nicole De Giorgi[2], Sebastiano Bellani[2,8] ✉, Aldo Di Carlo[1,6] ✉ & Francesco Bonaccorso [2,7] ✉

Perovskite solar cells promise to be part of the future portfolio of photovoltaic technologies, but their instability is slow down their commercialization. Major stability assessments have been recently achieved but reliable accelerated ageing tests on *beyond small-area cells* are still poor. Here, we report an industrial encapsulation process based on the lamination of highly viscoelastic semi-solid/highly viscous liquid adhesive atop the perovskite solar cells and modules. Our encapsulant reduces the thermomechanical stresses at the encapsulant/rear electrode interface. The addition of thermally conductive two-dimensional hexagonal boron nitride into the polymeric matrix improves the barrier and thermal management properties of the encapsulant. Without any edge sealant, encapsulated devices withstood multifaceted accelerated ageing tests, retaining >80% of their initial efficiency. Our encapsulation is applicable to the most established cell configurations (direct/inverted, mesoscopic/planar), even with temperature-sensitive materials, and extended to semi-transparent cells for building-integrated photovoltaics and Internet of Things systems.

Perovskite solar cells (PSCs) promise to revolutionize the photovoltaic (PV) industry thanks to power conversion efficiencies (PCEs) up to 26.1% and 33.9% in single-junction and tandem configurations[1–4], respectively, as well as their cost-effectiveness in terms of materials and high-throughput solution-manufacturing processes[5,6]. Nevertheless, to reach a Levelized Cost of Energy (LCoE) competing with those of market-dominating crystalline Si (c-Si) solar cells (>USD$0.05/ kWh)[7], long-term stability is still a challenge for PSCs[8], especially once

[1]CHOSE—Centre for Hybrid and Organic Solar Energy, University of Rome Tor Vergata, Via del Politecnico 1, 00133 Rome, Italy. [2]BeDimensional S.p.A., Via Lungotorrente Secca 30R, 16163 Genova, Italy. [3]Department of Mechanical, Energy, Management and Transport Engineering (DIME), Università di Genova, Genova, Italy. [4]Nanochemistry Department, Istituto Italiano di Tecnologia, Via Morego 30, 16163 Genova, Italy. [5]GreatCell Solar Italia SRL, Rome, Italy. [6]ISM-CNR, Istitute of Structure of Matter, Consiglio Nazionale delle Ricerche, Rome, Italy. [7]Graphene Labs, Istituto Italiano di Tecnologia, Via Morego 30, 16163 Genova, Italy. [8]These authors contributed equally: Paolo Mariani, Miguel Ángel Molina-García, Sebastiano Bellani. ✉e-mail: s.bellani@bedimensional.it; aldo.dicarlo@uniroma2.it; f.bonaccorso@bedimensional.it

assembled at module level, in which additional failure mechanisms, e.g., potential induced degradation and reverse bias effects, must be considered[8,9]. In general, the lifetime of a PSC is determined by both its intrinsic (e.g., polymorphism, defects, lattice strains and ion migration) and extrinsic (e.g., moisture, oxygen, heat, UV-light and reverse bias) factors[10,11]. Main degradation pathways include structural transitions and phase segregation of perovskite films or charge-transporting layers (CTLs)[12,13], often accompanied by morphological alterations[11,14]. Such mechanisms are frequently initiated by the migration of ions[15] and outgassing of volatile molecular species at perovskite grain boundaries and material interfaces[16]. These effects are commonly exacerbated in the presence of lattice defects[17]/strains[18] in the perovskite, as well as interfacial stresses resulting from the mismatches of lattices and thermal expansion coefficients (TECs) of the cell materials[19]. Several strategies, including compositional[20] and dimensional[21,22] engineering, defect passivation[23,24], grain boundary modification[25,26] and material interface engineering[27,28], have been proposed, and holistically combined to improve the intrinsic stability of PSCs[10,11,14,29]. Moisture and oxygen can react with perovskite absorbers, which finally decompose. By forming hydrogen bonds, water generates deprotonated organic cations, thus weakening the bond between the organic cation and the Pb-halide octahedral[30]. The proton can be then transferred to halide ions (e.g., $I^-$) via water molecules, producing volatile species (e.g., $CH_3NH_2$, HI and $PbI_2$). Meanwhile, oxygen can diffuse into the perovskite by occupying halide vacancies, and charged superoxides can form upon the photoexcitation of perovskites. These processes induce acid-base reactions with organic cations, which entail the formation of volatile species[31,32]. Importantly, the interplay between intrinsic and extrinsic factors ultimately determines the overall PSC stability[10]. In this context, elevate temperature/temperature variation[33], illumination and reverse bias[34] aggravate intrinsic degradation effects, being the light-sensitive perovskite absorbers subjected to photodissociation at elevated temperatures[15]. Encapsulation strategies have been consensually recognized as key in the realization of PSCs and corresponding modules lasting at least 20 years in outdoor conditions[10,35–37]. However, well-established encapsulation strategies reported for commercial PV technology cannot meet the distinctive requirements of PSCs, whose encapsulation concepts are still premature[10,35,38]. The main requirements that an encapsulant system should accomplish are[10,35,39]: 1) chemical inertness and chemical compatibility with underlying cell materials (e.g., no release of degrading chemicals, such as acetic acid and methacrylic acid for the case of ethylene vinyl acetate -EVA- and Surlyn, respectively); 2) low water vapor transmission rate (WVTR) ($\leq 10^{-4}$ g m$^{-2}$ day$^{-1}$) and oxygen transmission rate (OTR, $\leq 10^{-3}$ cm$^3$ m$^{-2}$ day$^{-1}$ atm$^{-1}$) to hinder the access of moisture and oxygen, while constraining the outgassing of volatile species[40]; 3) resistance to degradation processes (i.e., yellowing and release of degrading products for PSC materials) induced by UV radiation; 4) thermal stability up to 85 °C and low temperature ($\leq 120$ °C) processability to ensure compatibility with the thermal stability of perovskite and common CTLs; 5) optical transparency (i.e., transmittance $\geq 90\%$ from 400 to 1100 nm) for front side encapsulants; 6) electrically insulating properties (i.e., resistivity $>10^{13}$ Ω cm and high dielectric constant) to prevent leakage current and, hence, alleviate potential-induced degradation; and 7) mechanical properties, such as flexibility (i.e., low Young modules, preferably <20 MPa at 25 °C) and adhesivity (i.e., adhesion strength >0.1 MPa) to withstand thermo-mechanical stresses originated from daily temperature variation[36], as simulated by thermal cycling/shock ageing tests. So far, the combination of glass/pressure-tight polymer/glass encapsulation, in which the solar cell/module is sandwiched between two glass sheets using an encapsulant adhesive atop the PSC (blanket-cover approach), has been successfully applied for the realization of highly stable PSCs[10,35,36]. Among the adhesive encapsulant candidates, noteworthy examples are EVA, ionomers (Surlyn, Bynel and Jurasol), polysibutylene (PIB),

polyolefins (POEs), polyurethanes (PUs) and thermoplastic polyurethanes (TPUs). Also, edge sealants made of PIB-based butyl rubber tapes, UV-curable polymers, epoxy resins, silicones, and glass frits are commonly used to realize stable devices[41,42]. In particular, early reports demonstrated small-area PSCs passing accelerated ageing tests in compliance with international standards (i.e., international Electrotechnical Commission -IEC- 61215) and international Summit on Organic PV Stability (ISOS) protocols[10,35], e.g., damp heat ($\geq 1000$ h at 85 °C, relative humidity -RH- =85% for IEC 61215, ambient temperature for ISOS-D-2[43]; PCE retention >80%) and thermal cycling ($\geq 200$ temperature cycles between $-40$ °C and 85 °C for IEC 61215, minimum temperature $>-40$ °C for other lab procedures; PCE retention >80%)[33,36,40,44–51], as well as humidity freeze test[40]. However, such results have not yet been fully validated in perovskite solar modules (PSMs), remarking a commercialization gap between science and technology[52]. A recent study reported an effective encapsulation strategy based on a self-crosslinked fluorosilicone polymer gel, achieving non-destructive encapsulation at room temperature of both PSCs and a 25 cm$^2$ (active area = 15.8 cm$^2$)-PSM. By using an unspecified epoxy edge sealant, the encapsulated PSMs passed the IEC 61215 damp heat test (1000 h, PCE retention = 98%), but were not subjected to other relevant accelerated ageing tests[36]. Also, the corresponding encapsulated PSCs remarkably passed the IEC 61215 damp heat test (1000 h, PCE retention = 98%) and thermal cycling (220 cycles, PCE retention = 95%), even though continuous light soaking at 55 ± 5 °C for 1000 h led to a PCE drop of almost 20%. Overall, the encapsulation of PSMs by industrially compatible (high-throughput) methodologies has been rarely described[36,53], whereas ageing tests have not been systematically performed beyond direct outdoor performance evaluation[36,53]. For PSMs, the encapsulant materials must be compatible with a high throughput (minute-time scale) and cost-effective process[36,54]. Compared to traditional PV encapsulants, those for PSMs must also consider thermal management functionalities due to the low thermal conductivity of perovskite absorbers[36,55]. In addition, they must also have Pb-sequestrating abilities due to the risk of Pb release (~40 μg/kWh)[56] into the environment caused by the high solubility product constant ($K_{sp}$) of well-established perovskite by-products (i.e., $4.4 \times 10^{-9}$ M for $PbI_2$, which is 11 orders of magnitude higher than those of PbS and PbSe)[57,58]. Moreover, low-Young modulus encapsulants are typically recommended for PSCs and PSMs to avoid delamination issues caused by the mismatch of materials TECs[44]. Lastly, the encapsulant must not introduce cell-to-module losses caused by the large area of the edge seals, whose width around the glass edge should be less than 1 cm for square meter-sized solar panels. So far, most of the stability results have been demonstrated with encapsulant areas even larger than the photoactive area[38,40,44,45]. This means that such results must be still validated on practical device configurations ensuring market-attracting LCOEs[7].

In this work, we address the multifaceted challenges of encapsulants for PSMs by proposing an industrially compatible solvent- and strain-free encapsulation strategy based on a viscoelastic (semi-solid)/highly viscous (liquid) polyolefin, namely homopolymer PIB (not incorporating additives commonly used in PIB-based tapes, including butyl rubber edge sealants). By selecting a proper molecular weight of homopolymer PIB, the latter can exhibit a (highly viscoelastic) semi-solid-to-(highly viscous) liquid transition increasing the temperature from $-40$ °C to 85 °C, as those used to age PV devices through standardized tests. Polysibutylene is often reported as a common encapsulant material for PSCs. However, common PIB-based encapsulants contain several additives (e.g., isobutylene-isoprene co-polymer, silanes, tackifiers such as glycerol rosin ester, lamellar minerals such as talc and kaolin, metal oxides, carbon black and even molecular sieve desiccants) that enable the crosslinking of PIB chemically bond to surfaces, improve anti-ageing and impermeability properties, adjust the rheological/mechanical properties, and also control the esthetic

features (e.g., color)[54,59,60]. In this work, differently from commercially available PIB-based encapsulants commonly used in literature for PSCs[40,61,62], we propose low-molecular weight homopolymer PIB as a transparent (viscoelastic) semi-solid/(highly viscous) liquid processable in form of laminable films. The latter, herein deposited on glass substrates, can be used as primary encapsulant for PSCs via an industrially compatible solvent- and strain-free lamination protocols, aiming at solving limitations of current approaches based on solid encapsulants. In addition, we show that the adhesion, barrier and thermal management properties of our homopolymer PIB encapsulant can be improved by the addition of two-dimensional (2D) inorganic fillers, namely few-layer hexagonal boron nitride (h-BN) (nano)flakes produced at industrial scale through a patented wet-jet milling (WJM) exfoliation process of the native bulk powder[63–65]. After encapsulation (without extra edge sealing), our PSCs (either in mesoscopic and planar n-i-p configurations or inverted p-i-n configurations) and PSMs (mesoscopic n-i-p configurations), based on a perovskite chemistry well-established in large-area devices and at module level $(Cs_{0.08}FA_{0.80}MA_{0.12}Pb(I_{0.88}\,Br_{0.12})_3)$, retained more than 80% of the initial PCE after accelerated tests. These tests include thermal stress (ISOS-D-2 at 85 °C, >1000 h), light soaking (ISOS-L-1, >1000 h), customized thermal shock test (200 cycles between −40/+85 °C with abrupt temperature changes) and modified humidity freeze test (10 cycles with abrupt temperature changes between +85 °C and −40 °C and including a water immersion step before device freezing). Noteworthy, our customized/modified accelerated thermal shock and humidity freeze tests served as rapid ageing protocols to assess the moisture and temperature variation sensitivities of module designs[66–68], as targeted by more time-consuming IEC 61215 damp heat (1000 h) and thermal cycling (30-50 days) protocols, without the need to rely on expensive laboratory equipment (i.e., closed climatic chambers with RH control). Despite the superior encapsulant properties of PIB:h-BN encapsulants, homopolymer PIB was used as transparent encapsulant for semi-transparent PSCs, reaching a PCE of 6.8% and a bifaciality factor (defined as the ratio of PCEs measured with front and rear illuminations) as high as 89% after encapsulation. Overall, our results indicate that semi-solid/liquid encapsulation concepts efficiently mitigate either thermal and thermomechanical stresses during encapsulant application, while providing excellent barrier performance for the realization of long-term stable PSCs and PSMs, aiming at tackling the competition with Si-based PVs.

## Results

### Encapsulants characterization

Two types of encapsulants were prepared for the encapsulation of PSCs and PSMs, as described in the Methods section. Specifically, the first encapsulant is based on a room temperature highly viscous liquid transparent PIB with low-molecular weight (95,000), while the second one is an opaque composite of the same PIB and 2D h-BN (nano)flakes (hereafter named PIB:h-BN), being the latter produced by WJM exfoliation of bulk h-BN crystals[63,65,69]. Despite its amorphous and semi-solid/liquid nature (in the temperature range of −40/+85 °C), the homopolymer PIB herein selected exhibits a marked packing of its molecular chains, leading to high barrier properties[65,70]. Importantly, PIB has a resistivity on the order of $10^{16}\,\Omega$ cm, which is superior to that of EVA (ranging between $10^{13}$ and $10^{15}\,\Omega$ cm)[35], thus leading to potential induced degradation-suppressing properties[35,71]. In addition, previous studies have proved that the incorporation of 2D h-BN flakes into the PIB matrix is an effective strategy to enhance the barrier properties of pristine polymer against the permeation of water (and, thus, moisture) and other corrosive species[65,72]. The barrier properties of 2D h-BN are generally ascribed to its morphology with high-specific surface-area (1488 $m^2\,g^{-1}$ for monolayer h-BN)[73] and hydrophobic nature[74]. Moreover, the delocalized dense cloud of overlapping π-orbitals of h-BN represents a physical barrier against molecules or ions penetration,

leading to atomic impermeability[75]. Furthermore, 2D h-BN flakes exhibit a high thermal conductivity (e.g., >700 W $m^{-1}\,K^{-1}$ for monolayer h-BN and >100 W $m^{-1}\,K^{-1}$ for few-/multi-layer h-BN)[76,77], thus improving the thermal management properties of polymers when used as additives[36]. Inspired by our previous activities on anticorrosive coatings based on solid PIB with high molecular weight (800,000)[65], the barrier properties of the low-molecular weight semi-solid/liquid PIB proposed in this work were first tested through electrochemical methods. Figure 1a shows the potentiodynamic anodic polarization measurements and their corresponding Tafel analysis for representative PIB- and PIB:h-BN-coated steels (bare steel results are also shown for comparative purposes) immersed in a corrosive environment (3.5 wt.% NaCl water solution). These experiments were performed following the ASTM G5-14 standard (see Methods sections). The results evidence the barrier properties of PIB and PIB:h-BN films, which decrease the corrosion rate from 7.3 ×$10^{-1}$ mm year$^{-1}$ for structural steel to average values of $1.5 \times 10^{-1}$ mm year$^{-1}$ and $1.7 \times 10^{-4}$ mm year$^{-1}$ for the PIB- and PIB:h-BN-coated steels, respectively (Fig. 1b). Noteworthy, the addition of 2D h-BN flakes into PIB matrix improved the reproducibility of the anticorrosion performance of PIB films, even though both PIB and PIB:h-BN films lead to minimum corrosion rate as low as 5.6 × $10^{-4}$ mm year$^{-1}$ and $1.2 \times 10^{-5}$ mm year$^{-1}$. In general, the data proved the superior barrier properties of PIB:h-BN films (average corrosion inhibition efficiency = 99.97%) compared to homopolymer PIB (average corrosion inhibition efficiency ranging = 79.53%). The same trend was observed for homopolymer and composite films produced with solid (high-molecular weight) PIB films, as shown in Fig. S1 and ref. 65. This effect is attributed to the superior hydrophobicity resulting from the presence of 2D h-BN flakes (water contact angle of 88.3° ± 0.4° and 97.9° ± 0.4° for solid PIB and PIB:h-BN films, respectively, Fig. S2a–c), as shown in ref. 65. The adhesive properties of solid PIB and PIB:h-BN films were measured through pull-off tests following the ASTM D4541-02 standard, showing that the incorporation of 2D h-BN flakes into PIB increases the adhesive strength of the homopolymer PIB by 25% (Fig. S2d). Notably, both water contact angle and pull-off measurements were carried out on solid PIB and PIB:h-BN films since semi-solid/liquid films do not permit reliable measurements with these techniques. The WVTR of the proposed encapsulants was measured in a glass/pressure-tight polymer/glass system through calcium corrosion test (Ca test) (Fig. 1c, d), which analyzes the Ca corrosion through in-situ resistance measurements[78,79]. This sample configuration simulates the glass/pressure-tight polymer/glass encapsulation concept used for PSCs and PSMs in this work (blanket-cover approach), in which the moisture cannot pass through the glass (thus, its entrance occurs laterally from the device edge through the encapsulants)[80]. The calculated WVTR was ca. 2×$10^{-5}$ g $m^{-2}$ d$^{-1}$ for both systems based on semi-solid/liquid PIB and PIB:h-BN encapsulants (Fig. 1e). The UV-Vis transmittance spectra of the encapsulated Ca films remained almost unchanged over 15 days, thus excluding the Ca oxidation and, consequently, the ingress of moisture (Fig. 1f). The effects of 2D h-BN flakes on the thermal management properties were evaluated through infrared thermal imageing of glass/PIB/glass and glass/PIB:h-BN/glass systems (area = 5.6 cm×5.6 cm), which were realized through a lamination protocol resembling the one used for the encapsulation of PSCs and PSMs (see Methods section for details). The samples were heated up to 90 °C and then quickly transferred onto an Al platform at 25 °C. By means of an infrared camera, the maximum temperature of the samples was monitored over time. As shown in Fig. S3, the presence of 2D h-BN flakes improves the heat dissipation ability of the system compared to that based on bare PIB, reducing by 11.2% the time to reach 30 °C. Despite it was not possible to perform reliable thermal conductivity measurements of (viscoelastic) semi-solid/(highly viscous) liquid PIB because of the impossibility of realizing self-standing bulk objects with suitable thickness, a 2D h-BN flakes content of 5 wt% in other more solid polymeric matrix (as used in our PIB:h-BN) typically

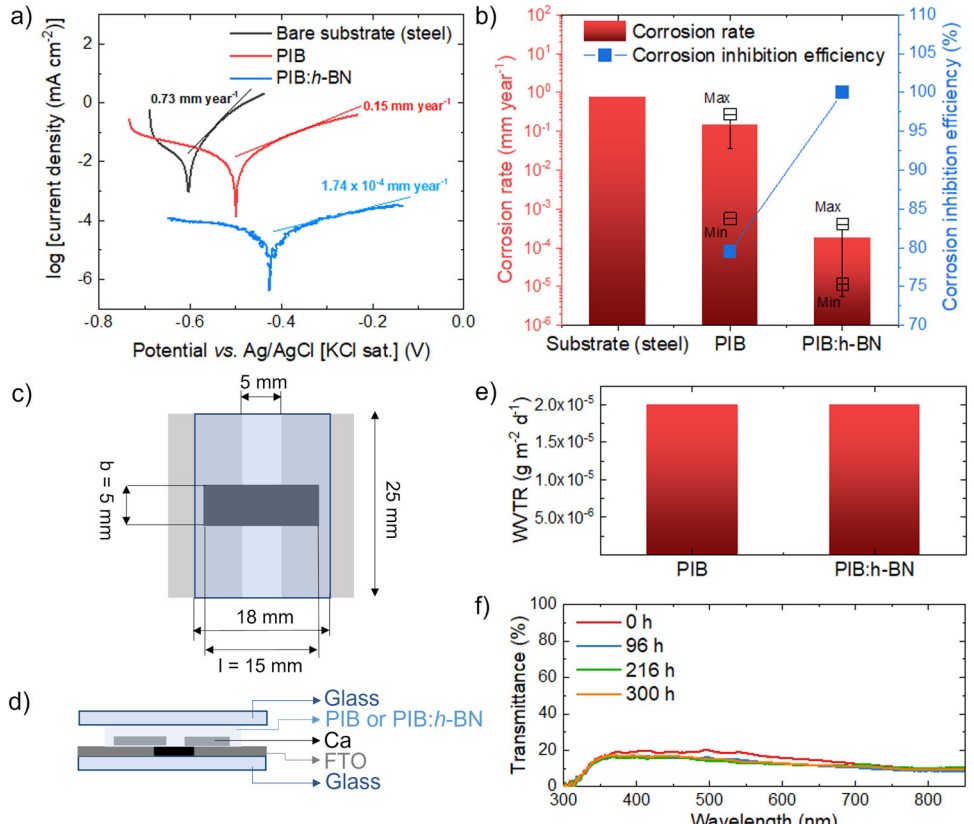

**Fig. 1 | Characterization of the barrier properties of the encapsulants. a** Anodic polarization curves (Tafel plots) of steel protected by PIB (low-molecular weight) and PIB:$h$-BN encapsulants (data acquired for the samples showing the highest corrosion rate among different replicas). The Tafel plot measured for bare steel is also shown for comparison. **b** Statistical analysis of the corrosion rates of the investigated systems and average corrosion inhibition efficiencies of the encapsulants. **c**, **d** Schematics (top-view and cross-section, respectively) of the sample configuration used for the Ca test. **e** WVTR measured through the encapsulants in the glass/pressure-tight polymer/glass encapsulation systems through the Ca test. **f** UV-Vis transmittance spectra of the samples measured through the Ca film at different times of environmental exposure.

improves significantly the thermal conductivity of the pristine polymer (e.g., by more than 80% in epoxy systems, as measured through Hot Disk measurements following ISO 22007-2 standard).

**Preliminary encapsulants assessment in PSCs**

The encapsulants were first tested in mesoscopic n-i-p PSC configurations (active area = 1 cm$^2$) based on Cs$_{0.08}$FA$_{0.80}$MA$_{0.12}$Pb(I$_{0.88}$Br$_{0.12}$)$_3$ perovskite and poly[bis(4-phenyl)(2,4,6-trimethylphenyl)amine] (PTAA) hole-transporting layer (HTL), following ISOS-D-2 (at 85 °C) and ISOS-L-1 protocols (after 240 h·ISOS-D-1 preconditioning)[43]. Noteworthy, PTAA was selected because of its superior thermal stability compared to other well-established HTLs, e.g., 2,2′,7,7′-tetrakis[N,N-di(4-methoxyphenyl)amino]−9,9′-spirobifluorene (spiro-OMeTAD), used for state-of-the-art performance PSCs[81]. Instead, the choice of this perovskite chemistry and the overall cell configuration relies on its high PCE proved at module/farm level by our group[53]. Prospectively, other perovskite chemistries, e.g., MA-free ones showing superior stability compared to MA-containing perovskites[40], could be also considered to further improve the stability results presented in this work. The cell structure was FTO/c-TiO$_2$/graphene-incorporating m-TiO$_2$/Cs$_{0.08}$FA$_{0.80}$MA$_{0.12}$Pb(I$_{0.88}$Br$_{0.12}$)$_3$/PEAI/PTAA/Au (Fig. 2a) (see Methods section for the definition of the acronyms). Graphene was incorporated into the m-TiO$_2$ to improve the electron extraction efficiency of the mesoscopic electron- transporting layer (ETL)[53,82–84], as well as to improve the stability of MA-based perovskites[85]. In addition, PEAI served as an ultrathin perovskite-passivating layer, as reported in previous studies[86,87]. Fig. 2b sketches the cell layout, which was designed to entirely cover the non-compact layers of the device with

the encapsulant, while using two flat metallic ribbons (commercial tape-like charge collectors) to bring the electrical contacts externally. The encapsulants were applied through an industrially compatible, high-throughput (total duration <45 min) lamination protocol (see details in the Methods section). Figure 2c shows a photograph of a representative device encapsulated with PIB:$h$-BN. Figure 2d shows the JV curves (reverse voltage scan) of a representative PSC, before and after encapsulation with PIB:$h$-BN. The JV curve recorded after 240 h-ISOS-D-1 test, performed before the ISOS-D-2 test, is also reported. Tables S1 and S2 list the PV parameters of the cells tested through ISOS-D-2 and ISOS-L-1 protocols, as extrapolated from their JV curves (for both reverse and forward voltage scan modes). The as-fabricated mesoscopic n-i-p PSCs based on PTAA HTLs show PCEs up to ~18.8%. Despite far from record certified PCE achieved by on small area active area (26.1% on 0.057127 cm$^2$)[3,4], our PCEs are significant for 1 cm$^2$-active area PSCs, whose record certified PCE of 21.6% remain unchallenged since 2019[88]. A recent article reported a record PCE of 24.35% for a 1.007 cm$^2$ cell by the NUS/SERIS group[4], but details are not disclosed in any accessible report[88]. As shown hereafter, either mesoscopic n-i-p configurations based on spiro-OMeTAD HTL and planar n-i-p configurations can improve further the PCE of our mesoscopic n-i-p PSCs based on PTAA HTLs, reaching maximum value of ~20.2% (for spiro-OMeTAD-based mesoscopic n-i-p PSCs), approaching further record certified PCE on large-area PSCs. For the encapsulated devices, the data indicate that the lamination process marginally affects the overall cell performances (absolute PCE drop <1%, regardless of the type of the encapsulant). During the ISOS-D-1 (i.e., dark storage at ambient temperature and ambient RH) test lasting 240 h the cells retained their

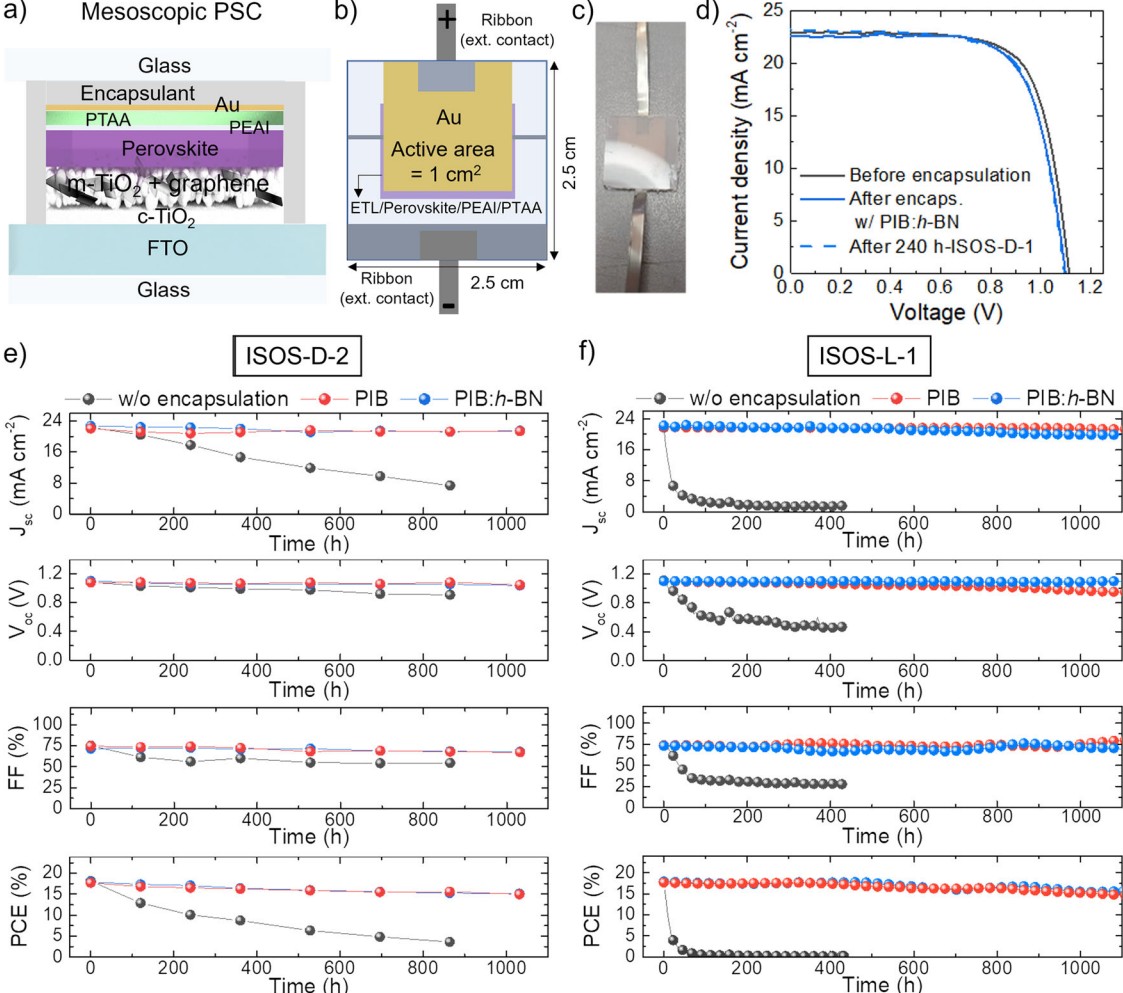

**Fig. 2 | Characterization of PSCs: ISOS-D-1/D-2 and ISOS-L-1 stability tests.**
**a** Sketch of the structure of the mesoscopic n-i-p PSCs based on $Cs_{0.08}$
$FA_{0.80}MA_{0.12}Pb(I_{0.88}Br_{0.12})_3$ perovskites and PTAA HTLs. **b** Schematic of the cell
layout (active area = 1 cm²), in which the non-compact layers of the device are fully
covered by the encapsulant. Two flat metallic ribbons are connected to the cell
terminals to bring the electrical contacts externally. **c** Photograph of a
representative mesoscopic PSC encapsulated with PIB:*h*-BN. **d** JV curves (reverse
voltage scan) measured for a representative mesoscopic PSC before and after
encapsulation with PIB:*h*-BN (before and after 240 h-ISOS-D1). **e**, **f** PV parameters of
the mesoscopic PSCs without encapsulation and with PIB and PIB:*h*-BN encapsu-
lants, acquired over >1000 h of ISOS-D-2 and ISOS-L-1 tests.

performances, confirming their shelf-life stability (absolute PCE drop
<1%). Afterwards, the cells underwent ISOS-D-2 and ISOS-L-1 tests,
sampling the cell PV parameters over >1000 h. As shown in Fig. 2e, f,
the unencapsulated cells quickly degraded during both the ageing
tests, showing a $T_{80}$ (defined as the time at which the PCE drops to 80%
of its initial magnitude; values estimated from a multi-order poly-
nomial fitting of the PCE data with $R^2 > 0.999$) < 70 h and <5 h for the
ISOS-D-2 and ISOS-L-1 tests, respectively. Contrary, the encapsulated
cells have shown $T_{80} > 1000$ h, regardless of the type of the encapsu-
lant. Noteworthy, no edge sealants were used during these experi-
ments, confirming the excellent barrier properties of our primary
encapsulants under PV operating conditions. Moreover, the stability of
the encapsulated cells was achieved in the presence of interconnection
ribbons, confirming the reliability of our encapsulants for practical PV
panels composed of ribbon-interconnected solar cells. Furthermore,
for the small-area cells investigated in this work, interconnect ribbons
may have negative effects on the overall lamination process. However,
the latter was successful thanks to the ability of the adopted highly
viscous liquid encapsulants to dissipate thermomechanical stresses,
leading to a strain-free encapsulation approach.

As mentioned above, our encapsulation approach was also pro-
bed on mesoscopic n-i-p PSCs based on spiro-OMeTAD, proving that

the proposed lamination protocol is compatible with more
temperature-sensitive HTLs compared to PTAA. As shown in Fig. S4,
Table S3, the lamination of PIB:*h*-BN encapsulant does not significantly
affect the cell performances (absolute PCE drop <1%), confirming the
results proved for PTAA-based mesoscopic n-i-p PSCs (see Fig. 2d).
Afterwards, 240 h-ISOS-D-1 test proved the shelf-life stability of the
investigated cells. Lastly, during the ISOS-D-2 test the unencapsulated
cell degraded, showing a $T_{80} < 240$ h, while the encapsulated cells have
shown $T_{80} > 1000$ h. In general, the hygroscopicity of lithium bis(tri-
fluoromethanesulfonyl) imide (LiTFSI) and the evaporation of 4-tert-
butylpyridine (tBP), used as spiro-OMeTAD dopants, promote moist-
ure entry into the cell structure and morphological changes of both
perovskite and spiro-OMeTAD HTL[81]. These effects are accelerated
with increasing temperature, leading to the formation of pinholes that
accelerate iodine migration to iodine-sensitive cellular components
(e.g., metal electrodes) and even cause connections between CTLs
(shunting pathways), leading to PCE losses[81]. In this context, our results
demonstrate that the combination of advanced PIB-based encapsu-
lants, which block moisture entry into PSCs, and ultrathin perovskite
passivation layers, e.g., PEAI, is a promising strategy to stabilize spiro-
OMeTAD-based PSCs operated at high temperature. In particular, PEAI
effectively passivate defects and trap states at the perovskite/spiro-

OMeTAD interface[89], helping our spiro-OMeTAD-based cell to withstand the lamination process at 90 °C for 10 min. However, our data indicate that proper encapsulants that effectively protects against the air/moisture ingress into the cell structure are crucial to avoid doped spiro-OMeTAD degradation through oxidation[90]. Our findings are well aligned with existing studies reporting excellent thermal stability of spiro-OMeTAD-based PSCs[91–93]. The effectiveness of PIB:$h$-BN encapsulants was further demonstrated on 1 cm²-active area planar n-i-p PSCs based on low-temperature processed SnO$_2$ as the ETL. As shown in Fig. S5 and Table S4, as fabricated cells exhibited a maximum PCE as high as 19.0%. After encapsulation with PIB:$h$-BN, they exhibited a T$_{80}$ of $ca.$ 2000 h during ISOS-D-2 test, whereas the unencapsulated ones have shown a lower T$_{80}$ of 700 h. Even though the use of SnO$_2$ ETL in planar PSCs may eliminate photo-induced degradation associated to TiO$_2$-based ETLs in mesoscopic structures[10,11], the reproducibility of planar PSCs at module level is still lower compared to mesoscopic configurations. The latter have been recently assembled with PTAA HTLs into a 4.5 m² stand-alone solar farm infrastructure[53,94] and, therefore, selected for the realization of the PSMs reported in this work hereafter. The universality of our encapsulation approach was also tested on 1 cm²-active area inverted p-i-n configurations based on PTAA as the HTL and [6,6]-phenyl-C$_{61}$-butyric acid methyl ester (PCBM) as the ETL. Long chain alkylammonium salt phenethyl ammonium chloride (PEACl) was used for perovskite surface treatment to simultaneously passivate the grain boundaries and the perovskite/PCBM interface[95]. As shown in Fig. S6 and Table S5, the cells retain their performance after encapsulation with PIB:$h$-BN, resulting in T$_{80}$ > 1000 h during ISOS-D-2 test, whereas the unencapsulated ones have shown T$_{80}$ < 360 h.

## Encapsulant validation in PSMs

The PIB and PIB:$h$-BN encapsulants were subsequently assessed in mesoscopic n-i-p PSMs (based on PTAA HTLs) consisting of 5 in series-connected cells with an active area of 2 cm² (total active area = 10 cm²), as reported in previous studies[96]. Fig. 3a shows the layout of the encapsulated mesoscopic n-i-p PSMs. The encapsulants were applied following the same lamination protocols reported for PSCs, paying attention to entirely covering the porous layers of the module structure. As for the case of the PSCs, no edge sealants were used in combination with our primary encapsulants. The PSMs were designed to avoid the need for ribbons to contact the positive and negative module terminals. Specifically, we printed two silver busbars along the edges of the modules exceeding the encapsulant. Thus, contrary to the case of PSCs, the thermomechanical stresses associated with the presence of ribbons (both during the encapsulation and during the operation of the PSM) have been eliminated. Figure 3b, c show the photographs of a representative PSM before (front and rear sides) and after encapsulation (rear side) with PIB:$h$-BN, respectively. Figure 3d shows the JV curves (reverse voltage scan) of two representative PSMs, before and after encapsulation with PIB and PIB:$h$-BN, respectively. The JV curves recorded after a 240 h-ISOS-D-1 test performed before the subsequent ageing tests are also reported. Tables S6 and S7 list the PV parameters of the modules tested through ISOS-D-2 and ISOS-L-1 tests, as extrapolated from their JV curves (for both reverse and forward voltage scan modes). Our PSMs reached a maximum PCE of 17%, which is relevant for module configurations proved at solar farm level[53] where the scalability and batch-to-batch reproducibility of the materials must be ensured together with high manufacturing yields. Prospectively, our encapsulant approach may also be assessed on more efficient PSMs configurations, now reaching record PCE up to 19.9% on 10 cm² active area[97]. As for the case of the cells, the encapsulation process marginally affected the overall module performances (absolute PCE drop <1%), which were also retained during 240 h-ISOS-D-1 tests (absolute PCE drop <1%) regardless of the type of the encapsulant. Afterwards, the unencapsulated PSMs quickly degraded during ISOS-D-2 and ISOS-L-1

tests (Fig. 3e, f), showing T$_{80}$ < 100 h and <3 h, respectively, resembling the instability observed for unencapsulated PSCs. Contrary, the encapsulated PSMs have shown T$_{80}$ > 1000 h.

To further assess the reliability of our encapsulants, as well as to specifically evaluate the barrier and thermal management functionalities of 2D $h$-BN flakes as encapsulant additives, encapsulated PSMs were subjected to two customized accelerated ageing stress, i.e., a thermal shock test (between -40/+85 °C) and, subsequently, a modified humidity freeze test. Figure 4a, b show the temperature profiles and environmental conditions (e.g., water immersion and air exposure) for our thermal shock and humidity freeze tests, while their comparison with IEC 61215 thermal cycling and humidity freeze tests is depicted in Fig. S7. Importantly, the cycle times of our tests (20 min for the thermal shock test and 25 min for the humidity freeze test) are significantly reduced compared to IEC 61215 tests minimum cycle times (>2.5 h for thermal cycling; >22 h for humidity freeze test). Thus, in our accelerated ageing stresses, abrupt temperature changes are supposed to induce severe thermomechanical stresses caused by the thermal expansion and contraction of materials with different TECs, critically jeopardizing the adhesion between various layers and the reliability of the electrical interconnections[98]. Previous studies in Si PERC cells proved that accelerated thermal cycling can represent an effective tool to rapidly prototype novel PV materials and module configurations while triggering degradation pathways that may not occur during traditional IEC 61215 thermal cycling[68]. Also, in our humidity freeze test, the water immersion step after heating the PSM to +85 °C and before freezing the PSM to -40 °C aims at replacing the long (>20 h) high-temperature (+85 °C) step at 85% RH required by IEC 61215 protocol to determine the ability of the module to withstand humidity penetration. Table S8 lists the PV parameters of the modules tested through customized thermal shock and humidity freeze tests, as extrapolated from their JV curves (for both reverse and forward voltage scan modes). As shown in Fig. 4c, the PSM encapsulated with PIB:$h$-BN withstood 200 thermal shock cycles, retaining 84.5% of the starting PCE. With the PIB encapsulant, the PSM retained 82.1% of the initial PCE after 200 cycles. These data indicate that the use of 2D $h$-BN flakes as thermally conductive additives in encapsulants is an effective strategy to improve the overall thermal management properties of PSMs, integrating passive cooling abilities into the encapsulant system. This is consistent with the thermal properties measured for our encapsulants (Fig. S3). After the thermal shock test, the PSMs encapsulated with PIB and PIB:$h$-BN were stressed further through the humidity freeze test (Fig. S8), retaining 72.1% and 86.0% of their PCE (before starting this test) after 10 cycles, respectively (Fig. 4d). Overall, PIB:$h$-BN slightly outperformed homopolymer PIB during thermal shock and humidity freeze tests, as expected by its distinctive barrier and thermal management properties (Fig. 1a, b, Figs. S2 and S3).

The effectiveness of the PIB:$h$-BN encapsulant to protect the PSMs from extrinsic factors was also assessed by measuring the Pb leakage of the encapsulated PSM immersed in water through inductively coupled plasma optical emission spectroscopy (ICP-OES) (Fig. S9). After water immersion the unencapsulated PSM rapidly degraded, showing yellowing associated with the decomposition of the perovskite to PbI$_2$. Because of its high solubility (340 mg L$^{-1}$, solubility product constant = $4.4 \times 10^{-9}$ M)[57,58] PbI$_2$ rapidly dissolved in water, causing cracking of the Au rear electrode. The detected Pb leakage (>60 μg cm$^{-2}$ after 24 h) is consistent with the Pb content in perovskite, typically between 0.1 and 1 g m$^{-2}$[56]. Contrary to unencapsulated devices, the perovskite in the encapsulated PSMs retained its starting color, preserving the perovskite phase. Consequently, the Pb leakage was drastically inhibited to values lower than 1 μg cm$^{-2}$ after 24 h (low Pb water contamination is likely associated with perovskite residuals nearby the encapsulant edges and not with the degradation of perovskite over the PSM active area). Similar Pb-leakage inhibition was observed for a PIB encapsulant protecting simple perovskite films.

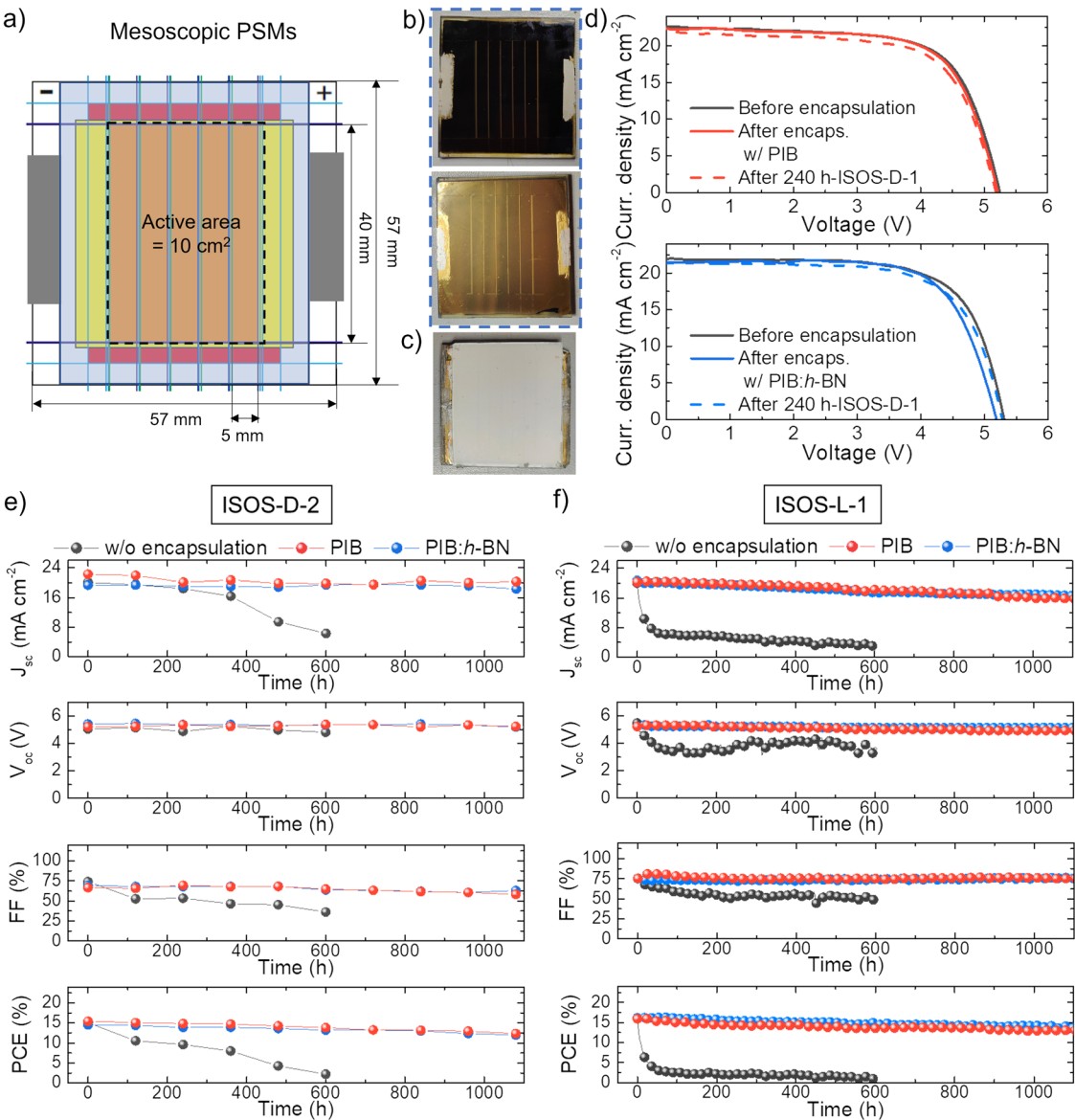

**Fig. 3 | Characterization of PSMs: ISOS-D-1/D-2 and ISOS-L-1 stability tests.**
**a** Schematic of the mesoscopic n-i-p PSM layout (cell active area = 2 cm²; total active area = 10 cm²), in which the non-compact layers are entirely covered by the encapsulant. **b** Photograph of a representative mesoscopic n-i-p PSM as fabricated (front and read sides: top and bottom picture, respectively) and **c** after encapsulation (rear side) with PIB:$h$-BN. **d** JV curves (reverse voltage scan) measured for the as-fabricated mesoscopic n-i-p PSMs before and after encapsulation with PIB (top panel) and PIB:$h$-BN (bottom panel) (before and after 240 h-ISOS-D1). **e**, **f** PV parameters of the PSMs without encapsulation and with PIB and PIB:$h$-BN encapsulants acquired over >1000 h of the ISOS-D-2 and ISOS-L-1 tests.

## Proof-of-concept semi-transparent PSCs based on PIB encapsulants

Even though PIB:$h$-BN encapsulants outperformed PIB ones during the most aggressive ageing tests (i.e., those involving abrupt temperature changes) reported for our PSMs, homopolymer PIB still has shown satisfactory performances, rarely achieved in literature without additional edge sealants. Consequently, transparent PIB encapsulants may find applications for high-PCE perovskite-based tandem systems[82,99,100] and building-integrated PVs –PIBVs– (e.g., smart windows, façades and agrivoltaics)[101]. Also, semi-transparent PV architectures, especially solution-processed ones (e.g., PSCs and organic solar cells), have attracted significant interest for indoor applications to power portable electronics and photonic devices for the Internet of Things (IoT), e.g., distributed sensors, remote actuators, and communication devices[102–105]. Here, indoor PVs may be subjected to less aggressive environmental conditions compared to outdoor PVs and, thus, PIB encapsulants may still be a suitable choice. To investigate these types

of applications, our PIB encapsulants were applied to semi-transparent PSCs based on a wide-bandgap (~2.3 eV)[106] FaPbBr₃ perovskite. Previously, the deposition of FaPbBr₃ perovskite (1.4 M) in a complete semi-transparent stack has been optimized by our group with both spin and blade coating technique reaching a maximum average visible transmittance (AVT) of 52% and a maximum bifaciality factor of 86.5%[107]. Here, we fabricated the FTO/c-TiO₂/FaPbBr₃/PTAA/ITO structure with a FaPbBr₃ solution of 1 M allowing to achieve an AVT value exceeding 60%. Figure 5a shows the UV-Vis transmittance spectra of a representative semi-transparent PSC before and after encapsulation with PIB. Interestingly, the AVT increased from 58.1% to 62.7% after encapsulation. Based on the reflectance spectra of the samples, this behavior is attributed to the decrease in reflection losses (i.e., improved matching of the refractive indices of the interface materials) after device encapsulation. The increase of the transmittance after PIB encapsulation is also observed for bare FTO, supporting our conclusion. The antireflective properties of PIB have been also confirmed by

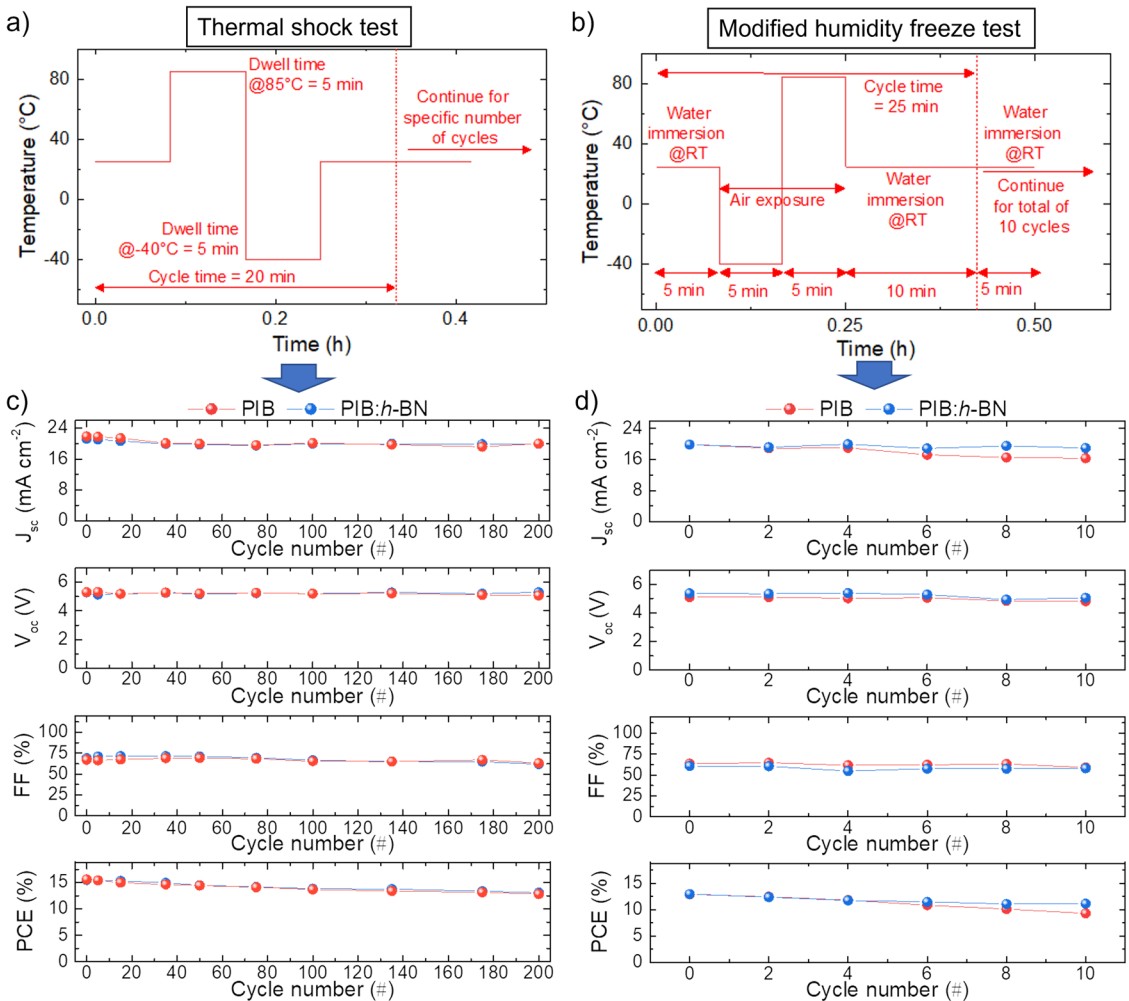

**Fig. 4 | Accelerated ageing tests for PSMs: thermal shock and humidity freeze tests. a** Temperature profile of the thermal shock test performed on the mesoscopic n-i-p PSMs encapsulated with PIB and PIB:*h*-BN. **b** Temperature profile and environmental exposure conditions of the humidity freeze test performed on the mesoscopic n-i-p PSMs encapsulated with PIB and PIB:*h*-BN. **c** PV parameters of the mesoscopic n-i-p PSMs encapsulated with PIB and PIB:*h*-BN acquired over >200 cycles of the thermal shock test. **d** PV parameters of the mesoscopic n-i-p PSMs encapsulated with PIB and PIB:*h*-BN acquired over >10 cycles of the customized humidity freeze test.

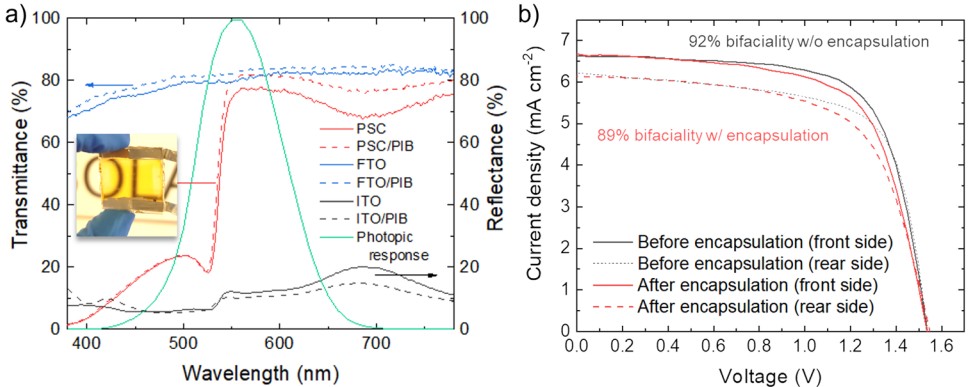

**Fig. 5 | Characterization of semi-transparent PSCs. a** UV-Vis transmittance spectra of a semi-transparent PSC before and after encapsulation with PIB (samples named PSC and PSC/PIB, respectively), bare FTO and FTO/PIB/glass (sample named FTO/PIB) (left y-axis). The reflectance spectra of ITO and ITO/PIB samples are also shown (right y-axis). The photograph of the semi-transparent PSCs is also shown. **b** JV curves measured for a representative semi-transparent PSC before and after encapsulation with PIB, for both front and rear side illuminations.

the reflectance spectra measured for ITO and PIB-coated ITO (ITO/PIB), also shown in Fig. 5a. Thus, PIB can acts as a kind of antireflective coating and, prospectively, future optical modeling and simulations (beyond the scope of this work) could be used to further reduce reflection losses by controlling the PIB thickness after the lamination process. Figure 5b shows the JV curves measured for representative semi-transparent PSCs before and after encapsulation for both front and rear side illuminations. Table S9 reports the PV parameters extracted from the JV curves, showing that the encapsulation processes almost retain the PCE of the unencapsulated devices. Significantly, the encapsulated cell exhibited a bifaciality factor of 89%, which is similar to the one measured before the encapsulation (92%).

## Discussion

In summary, we report a blanket-cover encapsulation approach for PSCs and PSMs based on the lamination of (highly viscoelastic) semi-solid/(highly viscous) liquid PIB-based adhesives atop the mesoporous cell layers. The viscoelasticity of PIB intrinsically limits the thermo-mechanical stresses caused by both the encapsulation process and the temperature gradients occurring during accelerated ageing stresses. The incorporation of 2D $h$-BN flakes into the PIB matrix improves the barrier and thermal management properties of homopolymer PIB, which, however, has demonstrated to be an optimal transparent encapsulant by itself for the realization of devices operating with reduced temperature fluctuation (e.g., indoor applications and specific BIPVs) compared to those taking place outdoor for conventional PVs. Without using any edge sealants the PSCs and PSMs encapsulated with our PIB-based encapsulants withstood multiple accelerated ageing tests, including ISOS-D1 preconditioning (240 h), ISOS-D2 (85 °C, >1000 h), ISOS-L1 (light soaking, >1000 h), as well as a customized thermal shock test (200 cycles) and modified humidity freeze test (10 cycles), retaining more than 80% of their initial (at the beginning of the test) PCE. Noteworthy, these results have been achieved with an MA-based perovskite, namely Cs0.08FA0.80MA0.12Pb(I0.88 Br0.12)3, and may, therefore, be further improved by using more stable MA-free perovskite chemistries. The combination of our semi-solid/liquid PIB-based encapsulants with advanced edge sealants can also be beneficial for the long-term stability and robustness of practical PSMs. Despite the superior encapsulant properties of PIB:$h$-BN compared to homopolymer PIB, the latter was used for the realization of proof-of-concept semi-transparent PSCs based on wide-bandgap FaPbBr$_3$. The encapsulated semi-transparent PSCs reached a PCE of 6.8% with a bifaciality factor as high as 89%, which is similar to the one measured before the encapsulation (92%). Prospectively, semi-transparent PSMs encapsulated with homopolymer PIB can be used for indoor applications (which entail conditions less harsh than outdoor ones), as well as for the front-side encapsulation of high-PCE perovskite-based tandem systems and BIPVs, possibly in combination with an edge sealant. The results reported in this study represent a breakthrough towards the realization of long-term stable PSMs through low-temperature and cost-effective semi-solid/liquid encapsulants combining high-throughput processability, perovskite/CTL compatibility, barrier properties, adhesivity, thermal and light-soaking stability, without the need to consider advanced perovskite chemistries with high internal stability or specific edge sealants.

## Methods
### Materials

TiO$_2$ (titanium dioxide) paste (30 NR-D), formamidinium iodide (FAI), methylammonium bromide (MABr), methylammonium chloride (MACl) and PEACl were purchased from GreatCell Solar. Lead(II) iodide (PbI$_2$), lead(II) bromide (PbBr$_2$), and cesium iodide (CsI) were purchased from TCI. Cesium bromide beads (CsBr), titanium(IV) isopropoxide (TTIP), diisopropoxytitanium bis(acetylacetonate) (Ti(AcAc)$_2$), acetyl acetone (AcAc), phenethylammonium iodide (PEAI), tris(2-(1H-pyrazol-1-yl)-4-

tert-butylpyridine)cobalt(III) tri[bis(trifluoromethane)sulfonimide] (FK209 Co(III) TFSI), bathocuproine (BCP) and copper beads (Cu beads), ethanol (EtOH) (anhydrous, ≥99.8 %), acetone (≥99.5%), acetonitrile (ACN) (≥99.8%), dimethylformamide (DMF) ≥ 99%), dimethyl sulfoxide (DMSO) (>99%), chlorobenzene (CB) (99.8%), 1,2-dichlorobenzene (DCB) (99%), toluene (>99.7%), N-methyl-2-pyrrolidone (NMP) (>97%), 2-propanol (IPA) (anhydrous, 99.5%), ethyl acetate (anhydrous, 99.8%), tBP and LiTFSI were purchased from Sigma-Aldrich. The SnO$_2$ dispersion in water (15%) was purchased from Alfa Aesar. Poly[bis(4-phenyl) (2,4,6-trimethylphenyl)amine] SOL2426M (average molecular weight, 10$^5$ kDa) was purchased from Solaris Chem. [6,6]-phenyl-C$_{61}$-butyric acid methyl ester was purchased from Solenne. Graphene dispersion in EtOH (0.9 mg mL$^{-1}$) was supplied by BeDimensional S.p.A. Room temperature highly viscous (Brookfield viscosity >100000 mPas at 10 rpm at temperature <120 °C) PIB (LMW-80, average molecular weight 95,000) was provided by TER Chemicals. Solid PIB (Oppanol® N80, average molecular weight 800,000) was purchased from BASF. Bulk powder of $h$-BN was supplied by Alfa Aesar. All the chemicals were used as received unless specified otherwise. Fluorine Tin Oxide (FTO)-coated glasses (sheet resistance R$_{SH}$ = 7 Ω sq$^{-1}$) were purchased from NSG-Pilkington. Indium-tin oxide (ITO) coated-glasses (R$_{SH}$ = 7 Ω sq$^{-1}$) were purchased from Kintec. Silver (Ag) paste 7713 was purchased from Dupont. Indium Tin Oxide (ITO) target for sputtering was purchased from TestBourne Ltd.

### Encapsulants preparation

To produce homopolymer PIB encapsulants, PIB was first dissolved in toluene with a PIB:toluene weight ratio of 1:1.5 and vigorously stirred (500 rpm) for 12 h at 800 rpm and 80 °C until a homogeneous solution was obtained. For the production of PIB:$h$-BN encapsulants, few-layer $h$-BN flakes were produced utilizing the patented WJM method by BeDimensional S.p.A[63,69]. Experimentally, a mixture of NMP and bulk $h$-BN with an NMP:$h$-BN weight ratio of 98:2 was pressurized into two jet streams, which collided in a nozzle (to produce the shear forces responsible for the exfoliation mechanism)[63,69]. The WJM-produced $h$-BN flakes dispersion was dried using a customized drier (BeDimensional S.p.A), ensuring solvent residuals less than 1 wt%, as detected by means of thermal gravimetric analysis (TGA) in N$_2$ atmosphere from 25 to 800 °C at a heating rate of 10 °C min$^{-1}$ using TGA Q500 (TA Instruments) thermogravimetric analyzer. Eventually, WJM-produced $h$-BN flakes were added into the PIB solution followed by mixing in a planetary centrifugal mixer (Thinky ARE-250 Mixing and Degassing Machine) at 1000 rpm for 5 min to produce $h$-BN/PIB composite resins with a $h$-BN weight percentage (wt.%) of 5% (excluding the solvent). The $h$-BN content was previously optimized through electrochemical tests compliant with ASTM G5-14, ASTM G59-97, ASTM G61-86, and ASTM G106-89 standards to maximize the barrier properties of the $h$-BN-incorporating encapsulants[65]. The electrochemical characterizations were carried out on samples produced either using (highly viscoelastic) semi-solid/(highly viscous) liquid PIB or solid (high-molecular weight) PIB. To produce the encapsulants, the PIB and PIB:$h$-BN resins were deposited by doctor blading onto 1 mm-thick glass substrates to be used for the glass/pressure-tight polymer/glass encapsulation of PSCs and PSMs. The resulting films were dried at room temperature for 1 h, followed by 15 h at 60 °C to evaporate the residual solvent.

### Encapsulants characterization

The thickness of the resulting homopolymer PIB or PIB:$h$-BN films was between 600 and 700 μm, as measured with a Trotec BB20 thickness measurement system (for these measurements, the encapsulant films were deposited on metallic substrates following the same procedure and parameters used for the glass coating). The water contact angle of the PIB and PIB:$h$-BN films was measured with an OSSILA L2004A1 contact angle goniometer, imageing a 10 μL water drop deposited on

the sample. To evaluate the barrier properties of the encapsulants, both electrochemical measurements and Ca tests were carried out. Electrochemical measurements were carried out using a BioLogic VMP3 Multichannel Potentiostat in a three-electrode 1 L electrochemical cell at room temperature in a 3.5 wt.% NaCl aqueous solution, following the procedures described in the ASTM G5-14 standard. A KCl-saturated Ag/AgCl radiometer Analytical REF201 Red Rod Reference Electrode (Biologic) was used as the reference electrode, whereas a graphite rod was used as the counter electrode. The standard working electrode assembly consisted of a cylindrical sample of structural steel substrate (S355) coated by PIB or PIB:$h$-BN films, drilled-and-tapped with a 3-48 UNC thread, and screwed onto the support rod. The PIB or PIB:$h$-BN films were produced by depositing the corresponding resins by doctor blading, followed by drying at room temperature for 1 h, and at 60 °C for 15 h to evaporate the residual solvent. The thickness of the PIB or PIB:$h$-BN films was ca. 60 μm, as measured with a Trotec BB20 thickness magnetic induction-based measurement system. A Teflon compression gasket ensured a leak-free seal. The open-circuit potential was monitored for 30 min, after which the corrosion performance of the coatings was investigated by potentiodynamic anodic polarization measurements and their Tafel analysis, as described in the ASTM G5-14 standard, for the determination of a metal's corrosion current ($i_{corr}$) and the corrosion potential[108,109]. The corrosion rate of the samples was calculated from $i_{corr}$ according to the Faraday law, i.e.: $CR = \frac{KW_{eq}i_{corr}}{D}$, where CR are the corrosion rate (in mm year$^{-1}$), K is a constant with a value of 3.27 × 10$^{-3}$, $W_{eq}$ is the equivalent weight of iron in ferrous compounds (27.9 g eq$^{-1}$), $i_{corr}$ is the corrosion current density (in μA cm$^{-2}$) and D is the density of steel (7.85 g cm$^{-3}$)[110]. The corrosion inhibition efficiency ($\eta_p$) of the composites was calculated from $i_{corr}$ by the following equation: $\eta_p\% = \frac{i^0_{corr} - i_{corr}}{i^0_{corr}} x 100$, where $i^0_{corr}$ and $i_{corr}$ are the corrosion current densities in the absence and presence of inhibitors, respectively[111].

Ca tests were performed on samples produced by depositing a Ca film on etched FTO-coated glass substrates with an area of 25 mm × 25 mm, which were then laminated on encapsulant-coated glass substrates using a heated press at ~100 °C to 150 °C in an N$_2$-filled glove box. Figure 1c, d depict the layout of the sample configuration used for the Ca tests. The FTO coatings were etched over a central strip of the substrate with an etched area of 25 mm × 5 mm. The Ca films were deposited by thermal evaporation with an area of 5 mm × 15 mm, which covered both parts of the FTO-etched region and FTO regions. The latter served as electrical contacts for the Ca films. The Ca films were covered by the encapsulants with an area of 25 mm × 18 mm. The WVTR through the encapsulants was measured through electrical (quantitative)[78] and optical (qualitative) analyzes of Ca corrosion[80]. The thermal management properties of the encapsulants were evaluated by monitoring the maximum temperature of glass/encapsulant/glass systems using a thermal camera (A655sc, FLIR) placed at ~50 cm from the sample surface. The samples were produced through a lamination protocol resembling the one used in this work for the encapsulation of PSCs and PSMs (see details hereafter). The samples were heated at 90 °C on a hot plate with a lid and then transferred to an Al platform at 25 °C. The temperature of the system was monitored during cooling. The thermal camera was controlled with FLIR's software (Temperature FLIR ResearchIR Max software), which was also used to process the temperature data. The adhesive properties of solid PIB and solid PIB:$h$-BN were measured through pull-off tests using an Instron 3365 dual-column dynamometer equipped with a 2 kN load cell and following the ASTM D4541-02 standard. The encapsulant resins were deposited on steel plates, which were clamped to the bottom anvil. Afterwards, the 15 mm diameter top piston was painted with cyanoacrylate adhesive and immediately put in contact with the sample. A force of 15 N was applied to the sample and the adhesive was let curing for 30 min. Normal displacement was then applied to the piston, with a rate of 1 mm min$^{-1}$ until separation.

## Perovskite solar cell and module fabrication

**Mesoscopic n-i-p PSCs.** Mesoscopic PSCs (active area = 1 cm$^2$) were fabricated on FTO-coated glass substrates cut in 2.5 × 2.5 cm$^2$ size. The FTO layer was patterned via laser etching using an Nd:YVO$_4$-pulsed UV laser system (BrightSolutions, Luce 40 laser), carrying out a P1 process to electrically separate the photoelectrode from the counter electrode. Then, the substrates were cleaned with brushing and without scratching the FTO surface, using a cleaning solution (Hellmanex) diluted with water (2:98 vol/vol). After this step, the substrates were sonicated in an ultrasonic bath first with acetone and then with IPA (10 min for each step). Afterwards, the substrates were air dried. Then, an UV/O$_3$-treated with a PSD Pro Series Digital UV Ozone System (Novascan) was used to remove organic contaminations. The c-TiO$_2$ layers were then deposited onto the patterned FTO by the spray pyrolysis of a dispersion of 0.16 M Ti(AcAc)$_2$ and 0.4 M AcAc in EtOH, setting the hot plate temperature at 465 °C and using air as the gas carrier at a pressure of 1.6 bar. The nozzle angle was about 45 °C with respect to the plane of the substrate, and the nozzle was moved with a serpentine path for 12-13 cycles (one every 10 seconds) until reaching a thickness of 50 nm, as measured by profilometry (Deektat Veeco 150). Then, the substrates were left for 15 min at 465 °C before slowly cooling them to room temperature. To fabricate the m-TiO$_2$ layers, the 30NR-D TiO$_2$ paste was diluted in anhydrous EtOH (1:5 w/w). After stirring overnight (>12 h), graphene was incorporated into the diluted 30NR-D TiO$_2$ paste by adding 1 vol% graphene dispersion in EtOH[53]. 120 μL of the graphene-incorporating paste was spin-coated onto c-TiO$_2$ layers at 3000 rpm for 30 s with an acceleration of 1500 rpm s$^{-1}$. The resulting layers were converted into graphene-incorporating m-TiO$_2$ through a multi-step sintering program: 1) 5 min-temperature ramp from room temperature to 150 °C, dwell time of 5 min; 2) 15 min-temperature ramp from 120 to 325 °C, dwell time of 5 min, 3) 5 min- temperature ramp from 325 to 375 °C, dwell time of 5 min; 4) 15 min-temperature ramp from 375 °C to 490 °C, dwell time of 30 min. To improve its wettability, the graphene-incorporating m-TiO$_2$ layers were UV-treated for 30 min with 5000-EC UV curing lamps (Dymax). Then, the samples were transferred to the glove box. The perovskite precursor solution was then prepared by dissolving FAI (1 M), PbI$_2$ (1.2 M), PbBr$_2$ (0.2 M), CsI (0.1 M), MABr (0.2 M) in DMF and DMSO (1:4 vol/vol) to obtain a perovskite composition of Cs$_{0.08}$ FA$_{0.80}$ MA$_{0.12}$ Pb (I$_{0.88}$ Br$_{0.12}$)$_3$. After stirring overnight, 90 μL of the perovskite precursor solution was spin-coated on the ETLs with a two-step protocol: 1) 2000 rpm for 10 s with an acceleration of 400 rpm s$^{-1}$; 2) 5000 rpm for 20 s with an acceleration of 2000 rpm s$^{-1}$. Subsequently, 100 μL of CB, used as antisolvent, was poured on the spinning substrates 15 s before the end of the spinning programme to induce the perovskite nucleation. After the perovskite deposition, the samples were annealed at 100 °C for 45 min to complete the crystal phase growth. An ultrathin perovskite-passivating PEAI layer was then deposited onto the perovskites by spin coating 100 μL of PEAI solution in IPA (concentration of 5 mg mL$^{-1}$) at 2500 rpm for 30 s, with an acceleration of 1250 rpm s$^{-1}$[86,87]. Then, the samples were annealed at 100 °C for 10 min. To deposit the HTL, a PTAA solution in toluene (10 mg mL$^{-1}$), doped with 7 μL mL$^{-1}$ of tBP and a 10 μl mL$^{-1}$ of a LiTFSI stock solution in ACN (170 mg mL$^{-1}$), was spin-coated at 3000 rpm for 20 s with an acceleration of 1500 rpm s$^{-1}$. As alternative to PTAA, spiro-OMeTAD was deposited by spinning 90 μL of spiro-OMeTAD solution in CB (73.5 mg mL$^{-1}$) doped with tBP (26.8 μL mL$^{-1}$), LiTFSI (16.6 μL mL$^{-1}$) (from stock solution in ACN (520 mg mL$^{-1}$) and FK209 Co(III) TFSI (7.2 μL mL$^{-1}$), at 4000 rpm for 1 min, with an acceleration of 1000 rpm s$^{-1}$. Finally, the mesoscopic n-i-p PSCs were completed by depositing 100 nm-thick Au back electrodes through thermal evaporation in a high vacuum chamber (10$^{-6}$ mbar).

**Planar n-i-p PSCs.** The planar PSCs (active area = 1 cm$^2$) were fabricated as the mesoscopic ones except for the ETL, which was produced by spin coating 100 μL of SnO$_2$ solution in water (15%) onto the FTO-coated glasses, previously UV-treated for 30 min to improve their

wettability. The $SnO_2$ solution was spun at 6000 rpm for 35 s with an acceleration of 3000 rpm s$^{-1}$. Then, the samples were annealed for 1 h at 150 °C to form a 70-80 nm-thick $SnO_2$ layer.

**Inverted p-i-n PSCs.** Inverted p-i-n PSCs were produced on ITO-coated glass substrates patterned with a Nd:YVO$_4$-pulsed UV laser system (BrightSolutions, Luce 40 laser) and then cut into 2.5 × 2.5 cm$^2$ samples. The ITO-patterned samples were cleaned in ultrasonic bath with a cleaning solution (Hellmanex) diluted with water (2:98 vol/vol), acetone and then isopropanol (15 min for each step). Any remaining solvent residual was blown off using air flow. UV-ozone treatment was then performed on the substrates for 15 min to remove all the residual organic contaminants, using a PSD Pro Series Digital UV Ozone System (Novascan). The samples were then transferred in a N$_2$-filled glove box and the PTAA (2 mg mL$^{-1}$ in toluene) was spin-coated at 5000 rpm for 20 s. The samples were annealed at 100 °C for 10 min. After cooling down, a film of PbI$_2$ and CsBr (with a ratio 10:1) was thermally co-evaporated onto the substrates. FAI (0.48 M), MABr (0.09 M) and MACl (0.09 M) were dissolved in EtOH and the solution was dynamically spin-coated on the substrates in a flow box filled with dry air (RH < 10%). The samples were then annealed in air (RH between 30 and 40%) at 150 °C for 15 minutes. On top of the perovskite layer, PEACl (1.5 mg mL$^{-1}$ in EtOH) was dynamically spin-coated at 4000 rpm and subsequently annealed at 100 °C for 10 minutes. Afterwards, PCBM (27 mg mL$^{-1}$ in CB:DCB−3:1 volume ratio) was spun at 1350 rpm for 20 s and annealed at 100 °C for 5 min. BCP (0.5 mg mL$^{-1}$ in IPA) was deposited at 2300 rpm for 20 s without any further drying. Finally, a Cu layer of 100 nm was thermally evaporated on top of the samples using a shadow mask.

**Semi-transparent PSCs.** The fabrication of semi-transparent PSCs (active area = 1 cm$^2$) started with the c-TiO$_2$ deposition onto FTO-coated glasses as for the case of mesoscopic PSCs. Before depositing the perovskite layers, the samples underwent a UV-light soaking for 10 min to improve their wettability. In an N$_2$-filled atmosphere, 80 μL of 1 M FaBr and 1 M PbBr$_2$ in DMSO solvent were spin-coated on the samples, pre-heated at 60 °C, at 4000 rpm for 20 s with an acceleration of 2000 rpm s$^{-1}$. After 10 s from the spin start, 200 μL of anhydrous ethyl acetate were dropped to induce the FaPbBr$_3$ perovskite crystallization. Subsequently, the c-TiO$_2$/FaPbBr$_3$ samples were annealed at 85 °C for 10 min. After cooling down the samples at room temperature, 90 μL of a PTAA solution in toluene (10 mg mL$^{-1}$), doped with 10 μL mL$^{-1}$ of tBP and 5 μL mL$^{-1}$ of a stock solution of LiTFSI (170 mg mL$^{-1}$ in ACN) were spin-coated with the same parameters used for the deposition of the perovskite layer. Afterwards, ITO was sputtered atop the samples at low temperature using an industrial magnetron sputtering (KENOSISTEC S.R.L., KS 400 In-Line) at $1.1 \times 10^{-3}$ mbar and 90 W RF power and purging inert Ar gas in the chamber at 40 sccm. By means of a sample holder, the samples were moved below the ITO target at 120 cm/min speed for 200 cycles to achieve a 200 nm-thick ITO top electrode with a sheet resistance of 25 Ω/sq, as measured by a four-probe unit installed in Arkeo Platform (Cicci Research S.r.l.).

**Mesoscopic PSMs**
The fabrication of mesoscopic PSMs followed the same step as the mesoscopic PSCs except for the quantities of solution poured onto the substrates and for the overall additional three laser ablation processes (P1-P2-P3, see Fig. S9) to define the series-connected layout with an active area of 10 cm$^2$ (single cell active area = 2 cm$^2$), as sketched in Fig. 3a[112]. The size of the FTO-coated substrates was 5.6 cm × 5.6 cm. The amounts of m-TiO$_2$ solution, perovskite precursor solution, chlorobenzene as antisolvent, PEAI solution, and PTAA solutions were 600 μL, 450 μL, 400 μL, 450 μL and 500 μL, respectively, for each module. The P1 process consisted of patterning the FTO-coated glass substrates to isolate 5 adjacent cells composing the final modules. The

width of P1 was 20 μm. The distance between the cells, cell width, and cell length were set to 0.5 mm, 5 mm, and 40 mm, respectively. After the cleaning step performed according to the procedure described for the mesoscopic PSCs, two Ag busbars (width 4 mm) were screen printed close to the edge of the substrate (2 mm from the edge), parallel to the longest size of the cell, using an automated screen printer (Baccini, Applied Materials). After the deposition, the busbars were dried onto a hot plate at 120 °C for 10 min. The busbars were then sintered during the spray pyrolysis deposition of c-TiO$_2$. After the deposition of the PTAA, P2 was carried out to clean the FTO interconnection areas. The width of P2 was 160 μm. After the thermal evaporation of the 100 nm-thick Au back electrodes, the adjacent cell isolation was accomplished with the P3 process. The width of P3 was 90 μm. The resolution of our laser ablation processes led to a geometrical fill factor of ~91%[113]. Table S10 lists the parameters used for laser ablation processes (P1, P2, and P3).

**Perovskite solar cells and modules lamination**
Both PSCs and PSMs were encapsulated through a multi-step, low-temperature (90 °C), differential pressure lamination process using an automatic two-chamber solar panel laminator (CORE – Model 2, Rise Technology srl), equipped with a cooling system to guarantee a high reproducibility on the lamination procedure and to reduce the exposure time of the materials to temperature causing their degradation. Experimentally, the entire surface of the devices was covered with the PIB or PIB:$h$-BN encapsulant-coated glass to ensure the glass/pressure-tight polymer/glass encapsulation of PSCs and PSMs (blanket-cover approach). The so-assembled laminates were put inside the laminator lower chamber, which exploited the differential pressure between the upper and lower chambers. After locking the laminator, 1) the laminator chambers were evacuated at moderate vacuum (pressure of ~1 mbar), while the laminates were heated from room temperature to 50 °C in 215 s (~7 °C min$^{-1}$). Then, 2) the upper chamber started to inflate to apply a final pressure of 30 mbar on the top of the substrates. The system required 500 s to stabilize such a low pressure in the laminator chamber. Then, 3) the temperature of the laminate was increased from 50 °C to 90 °C in 500 s (4.8 °C min$^{-1}$), which was then kept for 600 s. Afterwards, 4) the temperature of the laminate was decreased to 50 °C in 500 s (-4.8 °C min$^{-1}$). Lastly, 5) the pressure of the chambers returned to 1000 mbar in about 1 s and the laminator started opening while releasing the laminated devices.

**Device characterization**
J-V measurements of the devices were performed with a Class-A Sun Simulator (ABET 2000) equipped with an AM1.5 G filter (ABET). The sun simulator was calibrated to 1 Sun illumination condition with a Si-based reference cell (RR-226-O, RERA Solutions). Arkeo platform (Cicci Research S.r.l.) was used for J-V data acquisition under forward and reverse voltage scan modes, using a voltage step of 20 mV s$^{-1}$ and a voltage scan rate of 200 mV s$^{-1}$.

A UV-vis spectrophotometer (Shimadzu UV-2550) equipped with an integrated sphere was used for the acquisition of transmittance spectra of the encapsulant both for the optical characterization of the Ca test and for the characterization of semi-transparent devices. The sheet resistance of the Ca films during the calcium test was measured by the four-probe method using a Keithley 2620 source meter (Tektronix). The AVT values of semi-transparent PSCs were calculated according to the method reported in the ISO 9050:2003 standard using the following equation: $AVT = \left[ \int_{380}^{780} D(\lambda) \times T(\lambda) \times V(\lambda) d\lambda \right] / \left[ \int_{380}^{780} D(\lambda) \times V(\lambda) d\lambda \right]$, in which $D(y)$ is the incident light spectral distribution, $V(y)$ is the sensitivity factor of the human eye, and $T(\lambda)$ is the transmittance.

The ISOS-L-1 tests of the devices were performed in air and using an Arkeo-multichannel (Cicci Research S.r.l.) station based on 32 fully independent source meter units (+/− 10 V @ +/−250 mA) and an

ARKEO light soaker (VIS version) with low-mismatch LED-based system (400-750 nm). A standard Perturb & Observe tracking algorithm was selected for tracking the maximum power point of the devices, acquiring a J-V scan every 4 min. The ISOS-D-2 tests were carried out in a Lenton WHT4/30 oven (Hope Valley). A thermal shock test was performed by cycling the samples between −40 °C and +85 °C with abrupt temperature changes (from room temperature to +85 °C and vice versa, from room temperature to −40 °C and vice versa, see Fig. 4a). The modified humidity freeze test was performed after 200 thermal shock cycles and consisted of 10 thermal shock cycles between −40 °C and +85 °C, each cycle starting from a step of immersion in water at room temperature (see Fig. 4b). For the thermal shock and humidity freeze tests, a Lenton WHT4/30 oven and a low-temperature home-freezer were used to set the chamber temperature at +85 °C and −40 °C, respectively.

Inductively coupled plasma optical emission spectroscopy measurements were carried out on a ThermoFisher iCAP 7600 DUO Thermo spectrometer to measure the Pb leakage of the modules immersed in water, sampling the water solution at different times until a total immersion time of 24 h.

### Reporting summary

Further information on research design is available in the Nature Portfolio Reporting Summary linked to this article.

## Data availability

The source data generated in this study are provided in the Source Data file. Additional data that support the findings of this work are available from the corresponding authors upon request. Source data are provided with this paper.

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

## Acknowledgements

The work has been supported by the European Union's Horizon 2020 research and innovation programme under grant agreement number 881603—GrapheneCore3, European Union's Horizon Europe Framework Programme for research and innovation under grant agreement no. 101084124—DIAMOND, European Union's Horizon Europe Framework Programme for research and innovation under grant agreement no. 694101—2D-PRINTABLE. ADC gratefully acknowledges the support of the Mission Innovation grant between the Italian Ministry of Ecological Transition and ENEA (agreement 21A033302 GU n. 133/5-6-2021). FDG acknowledges funding from the European Union's Horizon 2020 research and innovation program under grant agreement no. 101006715—VIPERLAB. We are grateful to Prof. Liberato Manna for supporting this research with his research team.

## Author contributions

P.M. produced and characterized the PSCs and PSMSs, coordinating the experimental activities. E.M., S.P., L.V., F.D.G., and A.A. produced and characterized the PSCs and PSMs, formulating and characterizing device components. L.A.C. and F.D.G. contributed to the fabrication of

PSMs, engineering the P1-P2-3 laser patterning protocol and PSM design. S.B. and M.Á.M.-G. conceived the encapsulants concepts. M.Á.M.-G. and S.B. produced and characterized the encapsulants for the PSCs and PSMs coordinating the experimental activities. J.B. and F.M. produced and characterized the semi-transparent PSCs. M.I.Z., L.G., S.T., A.E.D.R.C., N.D.G, S.B., and F.B. produced and characterized 2D h-BN flakes, supporting the characterization of its polymeric composites. F.D. performed ICP-OES measurements. E.L. carried out the lamination of solar cells. P.M. and S.B. wrote the first draft and revised the manuscript with contributions from all the authors. A.D.C. and F.B. supervised and planned the experimental activities. All authors participated in the preparation of the final manuscript.

## Competing interests

F.B. is a co-founder and CSO, S.B., S.T and A.E.D.R.C are a senior scientists, M.A.M.-G., M.I.Z, L.G. and N.G are researchers at BeDimensional S.p.A., a company that is commercializing 2D materials. E.L. is a senior researcher of GreatCell Solar Italia SRL, a company commercializing solar-grade materials. The remaining authors declare no competing interests.
