## [Peer Review File · Nature Communications]

Low-temperature strain-free encapsulation for perovskite solar cells and modules passing multifaceted accelerated ageing testsReviewers' comments:

Reviewer #1 (Remarks to the Author):

Mariani et al. reported an industrially compatible encapsulation process that solves the problem of PSMs stability. Although PIB packaging materials have demonstrated effectiveness in improving stability, the use of PIB:h-BN packaging materials does not show significant improvements in stability, indicating that h-BN does not exhibit its advantages in this context. Furthermore, h-BN is widely used in the field of electronic packaging (J. Appl. Polym. Sci. 2023, 140, e53291; Mat. Today Phys. 2022, 22, 100594), so this method lacks novelty. The device efficiency is low (1cm² PCE=24.35% (Solar cell efficiency tables (version 62)), 10 cm² PCE≈20% (Adv. Mater. 2023, 2304625) in literature reports). The image layout is disorganized, specifically in Fig.2e-f, Fig.3b-f, Fig. 4d, Fig.S1a, and Fig.S4c. The stability of the packaged PSM has no obvious advantage among ISOS-D1 preconditioning (240 h), ISOS-D2 (85°C, >1000 h), ISOS-L1 (light soaking, >1000 h), as well as a customized thermal shock test (200 cycles) and modified humidity freeze test (10 cycles). Therefore, it is not recommended to publish in Nature Communications. Here are some concerns to address:

1. Although sufficient stability characterization has been conducted, the reviewers believe that potential mechanistic insights, especially at the atomic or molecular level, are insufficient.
2. The addition of hexagonal boron nitride (h-BN) improves the thermal management properties of the encapsulant. Unfortunately, there is not enough evidence to support this conclusion.
3. In the customized humidity freeze test, there is a lack of stability data for PIB-encapsulated devices.
4. The author proposes using transparent PIB homopolymer for the encapsulation of PSCs, achieving an improvement in visible transmittance (AVT). There is not enough evidence to prove the reasons for improvement, and the reviewers think this is unreasonable.
5. How universal is the encapsulation method? Can it be applied to the encapsulation of inverted PSCs?

Reviewer #2 (Remarks to the Author):

The manuscript titled "Strain-free semi-solid/liquid encapsulation for perovskite solar cells and modules passing thermal stress, light soaking, thermal shock and humidity freeze ageing tests" describes a reported method [Miguel Angel Molina-Garcia et al 2023 J. Phys. Mater. 6 035006], which uses the synthesized highly viscous polyolefin called PIB and hexagonal boron nitride (h-BN) flakes added PIB; these two kinds of coating materials were applied as the encapsulants in PSC and PSMs, and both gave outstanding performance in thermal stress thermal stress (ISOS-D-2 at 85°C, >1000 h), accelerated thermal shock and humidity freeze tests. Meanwhile, this work explored the novel encapsulant PIB in semi-transparent PSCs encapsulation.

Even though this work showed the good performance based on the PIB and h-BN added PIB, it seems that authors chose these materials based on their experience in industrial anticorrosive pigments, however, from the beginning of the material design, was not for perovskite solar cells. Thus, the encapsulation process involving heating lamination is only suitable for the non-temperature sensitive materials in PSCs such as PTAA in this work.

The state-of-the-art n-i-p structure PSCs rely on spiro-OMeTAD material, which can barely tolerate the temperature of 90°C for minutes based on the description in the method sections. The authors are suggested to provide the performance differences based on different PSCs structures/CTL materials, e.g., spiro-OMeTAD.

Besides, according to the details about PIB:h-BN preparation in SI, the NMP solvent was used to disperse h-BN. However the NMP is the well-use solvents for perovskite precursor, the authors are suggested to give the quantitative analysis of the residue of NMP in PIB:h-BN, and verify if the residue NMP can cause any negative effects on PSCs.

The PIB material are widely used in PSC encapsulation, see e.g., [Nature (2023). <https://doi.org/10.1038/s41586-023-06610-7>] [ACS Appl. Mater. Interfaces 2017, 9, 30, 25073–25081] [Science 368,eaba2412(2020)]. Please also compare the commercial PIB behavior with the synthesized PIB effects.

The last point is about the figures. The authors are encouraged to improve the figures to make the comparison clearer and easier to read.

On the basis of the above consideration, the current form of the manuscript is not suitable for Nature Communications. But if the authors can thoroughly address all the above points, it may be possible to reconsider it.

Reviewer #3 (Remarks to the Author):

The authors introduce an innovative encapsulation process for perovskite solar cells (PSCs) and perovskite solar modules (PSMs). This process involves the lamination of a highly viscoelastic semi-solid/viscous liquid encapsulant adhesive on the PSC/PSMs. Two types of encapsulants were prepared: a

transparent PIB and an opaque composite of PIB with two-dimensional hexagonal boron nitride (h-BN) flakes. The encapsulant's viscoelasticity inherently reduces thermomechanical stresses. Additionally, the inclusion of thermally conductive 2D h-BN flakes enhances the encapsulant's barrier properties against moisture corrosion. It also improved its thermal management properties. This is good result and can be published in Nature Communications.

I would suggest minor revision due to the comments below:

1. For Figure 4a left bottom corner, is the '-20 °C' right statement?
2. For the supporting tables S2-S6, why each condition has the statement 'after 240 h ISOS-D-1 test'?
3. For the module encapsulation part, since the encapsulant (PIB) will transform to a highly viscous liquid, will it get inside the scribing lines? If the answer is yes, how would this penetration affect the module stability?
4. The encapsulant preparation part mentioned about how to prepare the PIB with and without the h-BN. However, it doesn't talk about the volume ratio of the semi-solid PIB (MW 95000) and the solid PIB (MW 800000). Could the author further explain that and why is that ratio?
5. On page 5, the author mentioned that the PIB has several additives to make it crosslink during the heating process. It means that the PIB will have bonding with the device surface. Will this kind of chemical bonding be weakened or improved after thermal cycling? And will it have volume shrinkage or elongation after heating and cooling?
6. Did the author consider and test the oxygen permeability through the PIB?

We thank the Reviewers for their time and effort in evaluating our work on the development of novel type of encapsulants for perovskite-based photovoltaics (PVs). We note that Reviewers #2 and #3 reported a positive evaluation of our work, highlighting its key results. Furthermore, all the Reviewers provided valuable comments and recommendations. By addressing the Reviewers' requests, we do believe that we have overcome the previous Reviewer's concerns, while improving the overall quality of the manuscript, whose value can be now fully appreciated.

Below we report the point-to-point answer to the Reviewers' comments.

Reviewer #1

Mariani et al. reported an industrially compatible encapsulation process that solves the problem of PSMs stability.

We thank the Reviewer for his/her evaluation of our work. As shown hereafter, we did several efforts to perform supplementary measurements, replying to the Reviewers' comments point-by-point. We do hope that the Reviewer can now positively evaluate our work.

Although PIB packaging materials have demonstrated effectiveness in improving stability, the use of PIB:h-BN packaging materials does not show significant improvements in stability, indicating that h-BN does not exhibit its advantages in this context. Furthermore, h-BN is widely used in the field of electronic packaging (J. Appl. Polym. Sci. 2023, 140, e53291; Mat. Today Phys. 2022, 22, 100594), so this method lacks novelty.

To convince the Reviewer on the advancements reported by our work, we have included additional measurements, showing the functional role, *e.g.*, barrier and thermal management properties, of two-dimensional (2D) *h*-BN flakes when used additives for polymeric encapsulants, here specifically designed to be suitable for perovskite solar cells (PSCs). In particular, humidity freeze ageing test, needed to approve the commercialization of a new PV technology, revealed that, compared to homopolymer polyisobutylene (PIB), the superior thermal characteristics (heat dissipation ability, **Fig. S3**), as well as the superior hydrophobicity (**Fig. S2**) and barrier properties in corrosive electrochemical environments (**Fig. 1a**) of PIB:*h*-BN, improve the retention of the perovskite solar module (PSM) performances during humidity-freeze cycling. Nevertheless, we recognize that homopolymer PIB, here designed in terms of molecular weight/rheological characteristics, also represents a satisfactory encapsulant, enabling PSMs to retain more than 80% of their initial PCE after several accelerated ageing tests (ISOS-D-2, ISOS-L1 and thermal shock cycling). For this reason, homopolymer PIB was then tested for semi-transparent PSCs.

The text has been revised and now reads:

“After the thermal shock test, the PSMs encapsulated with PIB and PIB:h.BN were stressed further through the humidity freeze test (**Fig. S8**), retaining 72.1% and 86.0% of their PCE (before starting this test) after 10 cycles, respectively (**Fig. 4d**). Overall, PIB:*h*-BN slightly outperformed homopolymer PIB during thermal shock and humidity freeze tests, as expected by its distinctive barrier and thermal management properties (**Fig. 1a,b**, **Fig. S2** and **Fig. S3**).

Fig. 4. a) Temperature profile of the thermal shock test performed on the mesoscopic n-i-p PSMs encapsulated with PIB and PIB:h-BN. b) Temperature profile and environmental exposure conditions of the humidity freeze test performed on the mesoscopic n-i-p PSMs encapsulated with PIB and PIB:h-BN. c) PV parameters of the mesoscopic n-i-p PSMs encapsulated with PIB and PIB:h-BN acquired over >200 cycles of the thermal shock test. d) PV parameters of the mesoscopic n-i-p PSMs encapsulated with PIB and PIB:h-BN acquired over >10 cycles of the customized humidity freeze test.”

“Even though PIB:h-BN encapsulants outperformed PIB ones during the most aggressive ageing tests (*i.e.*, those involving abrupt temperature changes) reported for our PSMs, homopolymer PIB still have shown outstanding performances, rarely achieved in literature without additional edge sealants. Consequently, transparent PIB encapsulants may find applications for high-PCE perovskite-based tandem systems^{82,94,95} and building-integrated PVs –PIBVs– (*e.g.*, smart windows, façades and agrivoltaics).⁹⁶ In addition, semi-transparent PV architectures, especially solution-processed ones (*e.g.*, PSCs and organic solar cells), have attracted significant interest for indoor applications to power portable electronics and photonic devices for the Internet of Things, *e.g.*, distributed sensors, remote actuators, and communication devices.^{97–100} Here, indoor PVs

may be subjected to less aggressive environmental conditions compared to outdoor PVs, and thus PIB encapsulants may still be a suitable choice...”

As noticed by the Reviewer, *h*-BN has been already used as functional additive in polymeric composites because of its excellent barrier and thermal properties. In particular, in ref. *J. Appl. Polym. Sci.* **2023**, 140, e53291, as well as in most commercially available products, *h*-BN is used in its “bulk” form (<http://www.dcei.cn/en/info.aspx?c=6FE7C53C3CE96AF5&id=3476700CB72A4F00&a=s>).

For the sake of clarity, “bulk” refer to microplatelet structures which are different from the exfoliated forms consisting of (nano)flakes with atomic thickness, *i.e.*, flakes having less than 10 well-defined stacked material layers (according to ISO/TS 80004-13:2017 standard, this nomenclature has been defined for the specific case of graphene-based materials, but it is generally applied to layered crystals). The bulk form is used as precursor to produce 2D *h*-BN (nano)flakes, *i.e.*, the material form investigated in our work. As described in our manuscript, “bulk” *h*-BN powder is exfoliated through an innovative exfoliation technique, namely wet-jet milling exfoliation (WJM). This method is scalable and has been industrialized by BeDimensional S.p.A., currently commercializing few-layer graphene and *h*-BN flakes as 2D materials offering largest markets. So far, the massive deployment of 2D *h*-BN (nano)flakes is not established, since it is challenging to exfoliate bulk *h*-BN in massive amounts. Therefore, in these terms our work represents an important advancement towards the industrialization of 2D *h*-BN (nano)flakes in practical applications, including advanced encapsulants. The advantages of WJM exfoliation method compared to other exfoliating techniques, *e.g.*, shear mixing used in ref. *Mat. Today Phys.* **2022**, 22, 100594, is discussed in ref. *Mater. Horiz.* **2018**, 5, 890-904, as well as in the granted patent WO2017089987A1, currently licensed to BeDimensional S.p.A. Additionally, the advantages of 2D materials against of their counterparts are subject matter of extensive literature (*e.g.*, *Nanoscale*, **2015**, 7(11), 4598-4810). In the context of this work, 2D *h*-BN (nano)flakes can impart functional properties to the PIB at low content, *i.e.*, 2.5 wt%, clearly lower compared to those screened in other works using bulk *h*-BN (*e.g.*, 13.3 wt% in ref. *J. Appl. Polym. Sci.* **2023**, 140, e53291, where 26.67 wt% of Al₂O₃ particles were also included as additive into the polymeric matrix).

The device efficiency is low (1cm² PCE=24.35% (Solar cell efficiency tables (version 62)), 10 cm² PCE≈20% (Adv. Mater. 2023, 2304625) in literature reports).

Regarding the Reviewer’s comment on cell performances, we disagree that the performances of our PSCs are not relevant. First, in this work we mainly focused benchmark device configurations like those that have been validated by our groups in solar farm-scale infrastructures, *e.g.*, 4.5 m² stand-alone solar farm (see ref. *Nat. Energy* **2022**, 7, 597–607). A power conversion efficiency (PCE) of ~18.8% has been achieved on 1 cm²-active area mesoscopic n-i-p PSCs using poly[bis(4-phenyl)(2,4,6-trimethylphenyl)amine] (PTAA) as the hole-transporting layer (HTL). On planar n-i-p PSCs, we have instead achieved a PCE of ~19.0% on 1 cm² active area. In the revised manuscript, our PIB:*h*-BN encapsulants have been also validated on n-i-p PSCs based on Spiro-OMeTAD instead of PTAA. These cells reached a PCE of 20.2% on the same active area of 1 cm². In addition, we complemented our manuscript with the validation of our encapsulants on inverted p-i-n PSCs with 1 cm² active area, proving the universality of our approach.

Ref. *ACS Energy Lett.* 2023, 8, 9, 3829–3831 summarizes the record PCEs as certified in the NREL chart to represent the status (as of August 8, 2023) of all emerging solar cells, including PSCs. While PSCs achieved PCE of 26.1% on small area active area, namely 0.057127 cm², 1 cm² active area PSCs achieved record certified PCE of only 21.6%, moreover unchallenged since 2019. Thus, the PCE of our 1 cm²-active area PSCs (up to 20.2% for the new PSCs based on spiro-OMeTAD HTLs) is not so far from the state-of-the-art PCE measured on 1 cm². Noteworthy, our PCE results have been achieved without specific antireflective coatings (a common practice to boost the PCE during certification processes) and are accompanied with excellent device stability, as enabled using the encapsulant proposed in our work. Based on the “Solar cell efficiency tables (version 62)” by M. Green, the Reviewer refers to a PCE of 24.35% reached on ~1 cm². Nevertheless, this performance was associated to the ref. *Nat. Energy* 2022, 7(3), 229-237, which, however, reports tandem device with a PCE of 23.60% (certified PCE of 22.94%) achieved on an active area of 0.08 cm² and 21.77% on ~1 cm². To the best of our knowledge (as well as noticed in ref. *ACS Energy Lett.* 2023, 8, 9, 3829–3831), there are no reports/papers disclosing the details of the cells achieving the 24.35% PCE on 1 cm², even though declarations by NUS/SERIS group clarified that this result has been achieved on inverted p-i-n structures (<https://www.pv-magazine.com/2023/06/23/singapore-researchers-set-24-35-efficiency-record-for-perovskite-solar-cell/>).

The manuscript has been revised to summarize the above discussion, and now reads:

“The as-fabricated mesoscopic n-i-p PSCs based on PTAA HTLs show PCEs up to ~18.8%. Despite far from record certified PCE achieved by on small area active area (26.1% on 0.057127 cm²),^{3,4} our PCEs are significant for 1 cm²-active area PSCs, whose record certified PCE of 21.6% remain unchallenged since 2019.⁸⁸ A recent article reported a record PCE of 24.35% for a 1.007 cm² cell by the NUS/SERIS group,⁴ but details are not disclosed in any accessible report.⁸⁸ As shown hereafter, either mesoscopic n-i-p configurations based on spiro-OMeTAD HTL and planar n-i-p configurations can improve further the PCE of our mesoscopic n-i-p PSCs based on PTAA HTLs, reaching maximum value of ~20.2% (for spiro-OMeTAD-based mesoscopic n-i-p PSCs), approaching further record certified PCE on large-area PSCs.”

Lastly, regarding module performances, as noticed by the Reviewer, ref. *Adv. Mater.* 2023, 2304625 recently reported PSMs achieving PCE of 19.9% on 10 cm² active area, bridging the gap between PSM and PSC performances (PCE of 21.9% were obtained on small-active area (0.16 cm²) cells. Despite our PSMs have shown PCE lower compared to the record values reported in *Adv. Mater.* 2023, 2304625, we point out that our PSM configuration has been selected as benchmark to ensure reproducible performances, as we proved in relevant studies on the PSC upscaling up to solar farm-scale (*Nat. Energy* 2022, 7, 597–607). Overall, in accordance with the opinion recently expressed by Nat. Energy’s Reviewers and publishing editors, we consider that the performances of our PSMs are relevant for the community working on the upscaling of PSC technology.

The main text has been revised to discuss the above points, and now reads:

“Our PSMs reached maximum PCE of 17%, which are relevant for module configurations proved at solar farm level⁵³ where the scalability and batch-to-batch reproducibility of the materials must

be ensured together with high manufacturing yields. Prospectively, our encapsulant approach may be also assessed on more efficient PSMs configurations, now reaching record PCE up to 19.9% on 10 cm² active area.^{92*}

The image layout is disorganized, specifically in Fig.2e-f, Fig.3b-f, Fig. 4d, Fig.S1a, and Fig.S4c.

The layout of the figures has been improved, as also recommended by Reviewer #3.

The stability of the packaged PSM has no obvious advantage among ISOS-D1 preconditioning (240 h), ISOS-D2 (85°C, >1000 h), ISOS-L1 (light soaking, >1000 h), as well as a customized thermal shock test (200 cycles) and modified humidity freeze test (10 cycles). Therefore, it is not recommended to publish in Nature Communications.

Without encapsulation PSMs rapidly degraded during both ISOS-D-2 and ISOS-D-1, while the encapsulated PSMs demonstrated outstanding performance stability (without any edge sealants). For the sake of clarity, we report hereafter these results, shown in **Fig. 3** of the manuscript:

Fig. 3. a) Schematic of the mesoscopic n-i-p PSM layout (cell active area = 2 cm²; total active area = 10 cm²), in which the non-compact layers are entirely covered by the encapsulant. b) Photograph of a representative mesoscopic n-i-p PSM, as fabricated (front and rear sides: top and bottom picture, respectively) and c) after encapsulation (rear side) with PIB:h-BN. d) JV curves (reverse voltage scan) measured for the as-fabricated mesoscopic n-i-p PSMs before and after encapsulation with PIB (top panel) and PIB:h-BN (bottom panel) (before and after 240 h-ISOS-D1). e,f) PV parameters of the PSMs without encapsulation and with PIB and PIB:h-BN encapsulants acquired over >1000 h of the ISOS-D-2 and ISOS-L-1 tests.

Based on these excellent results, the encapsulated PSMs were subjected to additional ageing tests, *i.e.*, thermal shock cycling and humidity freeze tests. Overall, as highlighted in the abstract of the manuscript:

“Without any edge sealant, our encapsulated PSCs (including mesoscopic/planar n-i-p and inverted p-i-n configurations) and (mesoscopic n-i-p) PSMs, based on $\text{Cs}_{0.08}\text{FA}_{0.80}\text{MA}_{0.12}\text{Pb}(\text{I}_{0.88}\text{Br}_{0.12})_3$, withstood multifaceted accelerated ageing tests, retaining more than 80% of their initial power conversion efficiency (PCE). . .”

Here are some concerns to address:

1. Although sufficient stability characterization has been conducted, the reviewers believe that potential mechanistic insights, especially at the atomic or molecular level, are insufficient.

In this work, main efforts were devoted to design encapsulants meeting specific requirements for PSCs. As described in the main text and proper literature (*Energy Environ. Sci.* **2022**, 15, 13–55, *ACS Mater. Au* **2022**, 2, 215–236, *Renew. Sustain. Energy Rev.* **2021**, 151, 111608), these requirements are: 1) chemical inertness and chemical compatibility with underlying cell materials (*e.g.*, no release of degrading chemicals, such as acetic acid and methacrylic acid for the case of ethylene vinyl acetate -EVA- and Surlyn, respectively); 2) low water vapor transmission rate (WVTR) ($\leq 10^{-4}$ g m⁻² day⁻¹) and oxygen transmission rate (OTR) ($\leq 10^{-3}$ cm³ m⁻² day⁻¹ atm⁻¹) to hinder the access of moisture and oxygen, while constraining the outgassing of volatile species; 3) resistance to degradation processes (*i.e.*, yellowing and release of degrading products for PSC materials) induced by UV radiation; 4) thermal stability up to 85°C and low temperature ($\leq 120^\circ\text{C}$) processability to ensure compatibility with the thermal stability of perovskite and common CTLs; 5) optical transparency (*i.e.*, transmittance $\geq 90\%$ from 400 to 1100nm) for front side encapsulants; 6) electrically insulating properties (*i.e.*, resistivity $> 10^{13}$ Ω cm and high dielectric constant) to prevent leakage current and, hence, alleviate potential-induced degradation; and 7) mechanical properties, such as flexibility (*i.e.*, low Young modules, preferably < 20 MPa at 25°C) and adhesivity (*i.e.*, adhesion strength > 0.1 MPa) to withstand thermomechanical stresses originated from daily temperature variation, as simulated by thermal cycling/shock ageing tests.

All these requirements have been satisfied by the proposed encapsulants. Additional tests were carried out also on n-i-p PSCs more sensitive to the temperature, *i.e.*, PSCs based on spiro-OMeTAD HTLs, as well as on inverted p-i-n PSCs, proving the applications of our encapsulants to all relevant PSC configurations.

We also point out that the role of 2D *h*-BN (nano)flakes as polymeric additives has been deeply discussed in specialized literature, including mechanistic insights at the atomic/molecular level, as discussed in the Introduction section of our manuscript:

“In addition, previous studies have proved that the incorporation of 2D *h*-BN flakes into the PIB matrix is an effective strategy to enhance the barrier properties of pristine polymer against the permeation of water (and, thus, moisture) and other corrosive species.^{65,72} The barrier properties of 2D *h*-BN are generally ascribed to its morphology with high-specific surface-area (1488 m² g⁻¹ for monolayer *h*-BN)⁷³ and hydrophobic nature.⁷⁴ Moreover, the delocalised dense cloud of overlapping π -orbitals of *h*-BN represents a physical barrier against molecules or ions penetration,

leading to atomic impermeability.⁷⁵ Furthermore, 2D *h*-BN flakes exhibit a high thermal conductivity (e.g., $>700 \text{ W m}^{-1} \text{ K}^{-1}$ for *h*-BN monolayer and $>100 \text{ W m}^{-1} \text{ K}^{-1}$ for few-/multi-layer *h*-BN),^{76,77} thus improving the thermal management properties of polymers when used as additives.³⁶

2. The addition of hexagonal boron nitride (*h*-BN) improves the thermal management properties of the encapsulant. Unfortunately, there is not enough evidence to support this conclusion.

As shown in **Fig. S3**, the presence of 2D *h*-BN flakes improves the heat dissipation ability of the system compared to that based on bare PIB, reducing by 11.2% the time to reach 30°C. To strengthen our conclusion, we attempt to perform additional thermal conductivity measurements on PIB and PIB:*h*-BN by means of Hot Disk (Transient Plane Source -TPS-) method, following the ISO 22007-2 standard. Unfortunately, the liquid/semi-solid nature of our encapsulants impeded the realization of bulk samples needed for reliable hot disk measurements. Nevertheless, to convince the Reviewer, we report the analysis of the thermal conductivity as function of the percentage weight content (wt%) of the 2D *h*-BN flakes loading for another polymer:*h*-BN composite, namely epoxy:*h*-BN. As shown in **Fig. R1**, a 5 wt% of 2D *h*-BN (nano)flakes, as used in our PIB:*h*.BN encapsulants, increases the thermal conductivity by 87.8%.

Figure R1. Percentage increase of the thermal conductivity of an epoxy:*h*-BN composite as a function of the percentage weight content of 2D *h*-BN flakes. The thermal conductivity was measured with a Hot Disk Thermal Constants Analyzer (TPS 3500), according to the TPS method, as described in the ISO 22007-2 standard.

The main text now reads:

“The effects of 2D *h*-BN flakes on the thermal management properties were evaluated through infrared thermal imaging of glass/PIB/glass and glass/PIB:*h*-BN/glass systems (area = 5.6 cm×5.6 cm), which were realized through a lamination protocol resembling the one then used for the encapsulation of PSCs and PSMs (see Methods section for details). The samples were heated until 90°C, and then quickly transferred to an Al platform at 25°C. By means of an infrared camera,

the maximum temperature of the samples was monitored over time. As shown in **Fig. S3**, the presence of 2D *h*-BN flakes improves the heat dissipation ability of the system compared to that based on bare PIB, reducing by 11.2% the time to reach 30°C. Despite it was not possible to perform reliable thermal conductivity measurements of (viscoelastic) semi-solid/(highly viscous) liquid PIB because of the impossibility to realize self-standing bulk objects with suitable thickness, a 2D *h*-BN flakes content of 5 wt% (as used in our PIB:*h*-BN) embedded in other more solid polymeric matrices typically improves significantly the thermal conductivity of the pristine polymer (*e.g.*, by more than 80% in epoxy systems, as measured through Hot Disk measurements following ISO 22007-2 standard).”

In addition, humidity freeze test was also performed on the PSM encapsulated by PIB, after thermal shock cycling. The comparison between the results obtained with PIB and PIB:*h*-BN encapsulation reveals the superiority of the composite encapsulants to retain the PV performance of the module during humidity freeze cycling, because of the improvement of both barrier and thermal properties (**Fig. 1a, Fig. S2, Fig. S3**).

The main text has been revised to include the humidity freeze test of PSMs encapsulated with PIB, and now reads:

“As shown in **Fig. 4c**, the PSM encapsulated with PIB:*h*-BN withstood 200 thermal shock cycles, retaining 84.5% of the starting PCE. With the PIB encapsulant, the PSM retained 82.1% of the initial PCE after 200 cycles. These data indicate that the use of 2D *h*-BN flakes as thermally conductive additives in encapsulants is an effective strategy to improve the overall thermal management properties of PSMs, integrating passive cooling abilities into the encapsulant system. This is consistent with the thermal properties measured for our encapsulants (**Fig. S3**). After the thermal shock test, the PSMs encapsulated with PIB and PIB:*h*.BN were stressed further through the humidity freeze test (**Fig. S8**), retaining 72.1% and 86.0% of their starting PCE (before starting this test) after 10 cycles, respectively (**Fig. 4d**). Overall, PIB:*h*-BN slightly outperformed homopolymer PIB during thermal shock and humidity freeze tests, as expected by its distinctive barrier and thermal management properties (**Fig. 1a,b, Fig. S2 and Fig. S3**).

Fig. 4. a) Temperature profile of the thermal shock test performed on the mesoscopic n-i-p PSMs encapsulated with PIB and PIB:h-BN. b) Temperature profile and environmental exposure conditions of the humidity freeze test performed on the mesoscopic n-i-p PSMs encapsulated with PIB and PIB:h-BN. c) PV parameters of the mesoscopic n-i-p PSMs encapsulated with PIB and PIB:h-BN acquired over >200 cycles of the thermal shock test. d) PV parameters of the mesoscopic n-i-p PSMs encapsulated with PIB and PIB:h-BN acquired over >10 cycles of the customized humidity freeze test.”

3. In the customized humidity freeze test, there is a lack of stability data for PIB-encapsulated devices.

As discussed in the previous point, humidity freeze test on the PSM encapsulated by PIB has been included in the revised manuscript, demonstrating the superior performance of PIB:h-BN compared to PIB.

4. The author proposes using transparent PIB homopolymer for the encapsulation of PSCs, achieving an improvement in visible transmittance (AVT). There is not enough evidence to prove the reasons for improvement, and the reviewers think this is unreasonable.

The improvement of the visible transmittance (AVT) after encapsulation is mainly associated to the decrease of the reflectance compared to unencapsulated device. To prove the antireflection ability of the PIB, we have performed additional transmittance/reflectance measurements. As shown in the new **Fig. 5a**, the reflectance of the PIB-coated ITO (ITO/PIB) is reduced compared to bare ITO because of the better matching of the material refractive indexes. Consequently, the transmittance of encapsulated systems increased in the region of wavelengths between 530 nm to 780 nm, leading to the increase of the AVT.

The main text now reads:

“**Fig. 5a** shows the UV-Vis transmittance spectra of a representative semi-transparent PSC before and after encapsulation with PIB. Interestingly, the AVT increased from 58.1.8% to 62.7% after encapsulation. Based on the reflectance spectra of the samples, this behaviour is attributed to the decrease in reflection losses (*i.e.*, improved matching of the refractive indices of the interface materials) after device encapsulation. The increase of the transmittance after PIB encapsulation is also observed for bare FTO, supporting our conclusion. The antireflective properties of PIB have been also confirmed by the reflectance spectra measured for ITO and PIB-coated ITO (ITO/PIB), also shown in **Fig. 5a**. Thus, PIB can acts as a kind of antireflective coating, and, prospectively, future optical modelling and simulations (beyond the scope of this work) could be used to further reduce reflection losses by controlling the PIB thickness after the lamination process.”

Fig. 5. a) UV-Vis transmittance spectra of a semi-transparent PSC before and after encapsulation with PIB (samples named PSC and PSC/PIB, respectively), bare FTO and FTO/PIB/glass (sample named FTO/PIB) (left y-axis). The reflectance spectra of ITO and ITO/PIB samples are also shown (right y-axis). The photograph of the semi-transparent PSCs is also shown. b) JV curves measured for a representative semi-transparent PSC before and after encapsulation with PIB, for both front and rear side illuminations.”

5. How universal is the encapsulation method? Can it be applied to the encapsulation of inverted PSCs?

We have strongly appreciated this question, spurring us to perform additional tests on inverted (p-i-n) PSCs. As show hereafter, these new characterizations, as well as the ones on n-i-p PSCs

based on spiro-OMeTAD, indicate that our strain-free semi-solid/liquid low-temperature encapsulation methods is universal, *i.e.*, compatible with planar direct (n-i-p) and inverted (p-i-n), as well as mesoscopic n-i-p PSCs, based on either thermal-resistant HTL (*e.g.*, PTAA) or temperature-sensitive HTLs (*e.g.*, spiro-OMeTAD), and semi-transparent PSCs.

The main text now reads:

“The universality of our encapsulation approach was also tested on 1 cm²-active area inverted p-i-n configurations based on PTAA as the HTL and [6,6]-phenyl-C₆₁-butyric acid methyl ester (PCBM) as the ETL. Long chain alkylammonium salt phenethyl ammonium chloride (PEACl) was used for perovskite surface treatment to simultaneously passivate the grain boundaries and the perovskite/PCBM interface.⁹⁰ As shown in **Fig. S6** and **Table S5**, the cells retain their performance after encapsulation with PIB:*h*-BN, resulting in T₈₀ > 1000 h during ISOS-D-2 test, whereas the unencapsulated ones have shown T₈₀ < 360 h.”

Fig. S6. a) Sketch of the structure of the large-area (1 cm²) inverted p-i-n PSCs based on PTAA HTLs and PCBM ETLs. b) JV curves measured for the as-fabricated inverted p-i-n PSCs before and after encapsulation with PIB:*h*-BN (before and after 240 h-ISOS-D-1). c) PV parameters of the investigated inverted p-i-n PSCs acquired over >2000 h of ISOS-D-2 test.

Table S5. PV parameters of the large-area (1 cm²) inverted p-i-i PSCs based PTAA HTLs and PCBM ETLs, classified according to their encapsulants, tested through ISOS-D-2 protocols (after 240 h ISOS-D-1 test).

Encapsulant type	Status	Voltage scan mode	Voc (V)	FF (%)	Jsc (mA cm ⁻²)	PCE (%)
w/o encapsulation (reference)	As-fabricated	Reverse	1.08	68.7	22.0	16.3
		Forward	1.06	69.3	22.1	16.3
	After 240 h ISOS-D-1	Reverse	1.07	68.3	22.0	16.1
		Forward	1.07	68.1	22.0	16.0
PIB:h-BN	As-fabricated and before encapsulation	Reverse	1.08	69.2	22.1	16.6
		Forward	1.07	69.9	21.9	16.4
	After encapsulation	Reverse	1.07	69.6	21.9	16.3
		Forward	1.06	69.6	21.9	16.3
	After 240 h ISOS-D-1	Reverse	1.06	69.6	21.9	16.1
		Forward	1.06	69.1	22.0	16.0

Reviewer #2

The manuscript titled “Strain-free semi-solid/liquid encapsulation for perovskite solar cells and modules passing thermal stress, light soaking, thermal shock and humidity freeze ageing tests” describes a reported method [Miguel Angel Molina-Garcia et al 2023 J. Phys. Mater. 6 035006], which uses the synthesized highly viscous polyolefin called PIB and hexagonal boron nitride (h-BN) flakes added PIB; these two kinds of coating materials were applied as the encapsulants in PSC and PSMs, and both gave outstanding performance in thermal stress thermal stress (ISOS-D-2 at 85°C, >1000 h), accelerated thermal shock and humidity freeze tests. Meanwhile, this work explored the novel encapsulant PIB in semi-transparent PSCs encapsulation.

We thank the Reviewer for his/her evaluation of our manuscript.

Even through this work showed the good performance based on the PIB and h-BN added PIB, it seems that authors chose these materials based on their experience in industrial anticorrosive pigments, however, from the beginning of the material design, was not for perovskite solar cells. Thus, the encapsulation process involving heating lamination is only suitable for the non-temperature sensitive materials in PSCs such as PTAA in this work. The state-of-the-art n-i-p structure PSCs rely on spiro-OMeTAD material, which can barely tolerate the temperature of 90oC for minutes based on the description in the method sections. The authors are suggested to provide the performance differences based on different PSCs structures/CTL materials, e.g., spiro-OMeTAD.

We thank the Reviewer for this important observation, which deserve additional clarifications from our side. As noticed by the Reviewer, the development of the proposed encapsulants has

been inspired by our knowledge on anticorrosion coatings in building, construction and maritime sector. Part of this knowledge is discussed in our previous work *J. Phys. Mater.* **2023**, 6, 035006. However, in the current work the PIB-based coatings have been significantly re-designed compared to our previous studies for the specific case of PSCs. In particular, by reducing the molecular weight of the homopolymer PIB we attained the so-called **strain-free semi-solid/liquid low-temperature** encapsulation methods, which was successful for the encapsulation of PSCs and PSMs, particularly sensitive to either thermal or mechanical stresses. More in detail, in the proposed encapsulants, low molecular weight (95,000) PIB (LMW-80, supplied by TER Chemical) was used, while in our previous work solid high-molecular weight (800,000) PIB (Oppanol N80, purchased from BASF) was considered as a better choice as anticorrosion coatings for the protection of structural steel from the corrosion occurring in seawater conditions.

We agree with the Reviewers that these aspects were not clear in our manuscript, which has been revised and now reads:

“In this work, differently from commercially available PIB-based encapsulants, we propose low-molecular weight homopolymer PIB as a transparent (viscoelastic) semi-solid/(highly viscous) liquid processable in form of laminable films. The latter, herein deposited on glass substrates, can be used as primary encapsulants for PSCs *via* an industrially compatible solvent- and strain-free lamination protocols.”

“Two types of encapsulants were prepared for the encapsulation of PSCs and PSMs, as described in the Methods section. Specifically, the first encapsulant is based on a room temperature highly viscous liquid transparent PIB with low-molecular weight (95,000), while the second one is an opaque composite of the same PIB and 2D *h*-BN (nano)flakes (hereafter named PIB:*h*-BN), being the latter produced by wet-jet milling exfoliation of bulk *h*-BN crystals.^{63,65,69} ...

...Inspired by our previous activities on anticorrosive coatings based on solid PIB with high molecular weight (800,000),⁶³ the barrier properties of the low-molecular weight semi-solid/liquid PIB proposed in this work were first tests through electrochemical methods.”

As recommended by the Reviewer, to prove that the universality of our encapsulation approach, additional tests were performed on mesoscopic n-i-p PSCs based on spiro-OMeTAD as the HTL instead of the PTAA. These results have been included in the revised manuscript, which now reads:

“As mentioned above, our encapsulation approach was also probed on mesoscopic n-i-p PSCs based on spiro-OMeTAD, proving that the proposed lamination protocol is compatible with more temperature-sensitive HTLs compared to PTAA. As shown in **Fig. S4, Table S3**, the lamination of PIB:*h*-BN encapsulant does not significantly affect the cell performances (absolute PCE drop <1%), confirming the results proved for PTAA-based mesoscopic n-i-p PSCs (see **Fig. 2d**). Afterwards, 240 h-ISOS-D-1 test proved the shelf-life stability of the investigated cells. Lastly, during the ISOS-D-2 test, the unencapsulated degraded, showing a $T_{80} < 240$ h, while the encapsulated cells have shown $T_{80} > 1000$ h.”

Fig. S4. a) Sketch of the structure of the large-area (1 cm^2) mesoscopic n-i-p PSCs based on $\text{Cs}_{0.08}\text{FA}_{0.80}\text{MA}_{0.12}\text{Pb}(\text{I}_{0.88}\text{Br}_{0.12})_3$ perovskites and spiro-OMeTAD HTLs. b) JV curves measured for the as-fabricated mesoscopic n-i-p PSCs based on spiro-OMeTAD HTLs before and after encapsulation with PIB:h-BN (before and after 240 h-ISOS-D-1). c) PV parameters of the

Table S3. PV parameters of the large-area (1 cm^2) mesoscopic n-i-p PSCs based on spiro-OMeTAD HTLs, classified according to their encapsulants, tested through ISOS-D-2 protocols (after 240 h ISOS-D-1 test).

Encapsulant type	Status	Voltage scan mode	Voc (V)	FF (%)	Jsc (mA cm^{-2})	PCE (%)
w/o encapsulation (reference)	As-fabricated	Reverse	1.11	74.8	23.8	19.70
		Forward	1.10	73.7	23.7	19.29
	After 240 h ISOS-D-1	Reverse	1.11	73.2	23.6	19.17
		Forward	1.10	72.8	23.9	19.10
PIB:h-BN	As-fabricated and before encapsulation	Reverse	1.11	76.9	23.6	20.16
		Forward	1.11	74.8	23.9	19.77
	After encapsulation	Reverse	1.10	75.1	23.4	19.26

		Forward	1.09	73.9	23.8	19.13
	After 240 h ISOS-D-1	Reverse	1.10	73.5	23.6	19.11
		Forward	1.09	73.7	23.7	19.06

Besides, according to the details about PIB:h-BN preparation in SI, the NMP solvent was used to disperse h-BN. However the NMP is the well-use solvents for perovskite precursor, the authors are suggested to give the quantitative analysis of the residue of NMP in PIB:h-BN, and verify if the residue NMP can cause any negative effects on PSCs.

We thank the Reviewer to point out this important aspect. As noticed by the Reviewer, NMP was used to exfoliate “bulk h-BN” in form of 2D h-BN flakes by means of BeDimensional’s proprietary liquid-phase exfoliation method, namely wet-jet milling (WJM). BeDimensional Quality & Control department guarantees that 2D h-BN batches contain less than 1% by weight of NMP residues by thermogravimetric analysis (TGA). This information was specified in the Methods section of the revised manuscript. Solvent residues can be further eliminated during drying of encapsulants after being deposited on glass substrates from their solvent-borne resin in toluene.

The main text now reads:

“Experimentally, a mixture of NMP and bulk h-BN with an NMP:h-BN weight ratio of 98:2 was pressurized into two jet streams, which collided in a nozzle (to produce the shear forces responsible for the exfoliation mechanism).^{63,69} The WJM-produced h-BN flakes dispersion was dried using a customized drier (BeDimensional S.p.A), ensuring a solvent residuals less than 1 wt%, as detected by means of thermal gravimetric analysis (TGA) in N₂ atmosphere from 25 to 800°C at a heating rate of 10°C min⁻¹ using TGA Q500 (TA Instruments) thermogravimetric analyzer.”

For the sake of the clarity, we report hereafter for this reviewer the TGA curves (**Fig. R2**) measured for the 2D h-BN batch used in this work:

Fig. R2. TGA curve measured for 2D *h*-BN batch used in this work, showing marginal NMP residues (< 1wt%).

The PIB material are widely used in PSC encapsulation, see e.g., [Nature (2023). <https://doi.org/10.1038/s41586-023-06610-7>] [ACS Appl. Mater. Interfaces 2017, 9, 30, 25073–25081] [Science 368,eaba2412(2020)]. Please also compare the commercial PIB behavior with the synthesized PIB effects.

We have appreciated this Reviewer’s comment, and we have revised our manuscript to highlight the difference between our and commercially available PIB-based encapsulants, as those reported in the relevant literature mentioned by the Reviewers and now cited in our manuscript.

The main text now reads:

“In this work, we address the multifaceted challenges of encapsulants for PSMs by proposing an industrially compatible solvent- and strain-free encapsulation strategy based on a viscoelastic (semi-solid)/highly viscous (liquid) polyolefin, namely homopolymer PIB (not incorporating additives commonly used in PIB-based tapes, including butyl rubber edge sealants). By selecting a proper molecular weight of homopolymer PIB, the latter can exhibit a (highly viscoelastic) semi-solid-to-(highly viscous) liquid transition increasing the temperature from -40°C to 85°C, as those used to age PV devices through standardized tests. PIB is often reported as a common encapsulant material for PSCs. However, common PIB-based encapsulants contain several additives (e.g., isobutylene-isoprene co-polymer, silanes, tackifiers such as glycerol rosin ester, lamellar minerals such as talc and kaolin, metal oxides, carbon black and even molecular sieve desiccants), which enable the crosslinking of PIB, chemically bond to surfaces, improve anti-ageing and impermeability properties, adjust the rheological/mechanical properties, and also control the aesthetic features (e.g., colour).^{54,59,60} In this work, differently from commercially available PIB-based encapsulants commonly used in literature for PSCs,^{40,61,62} we propose low-molecular weight homopolymer PIB as a transparent (viscoelastic) semi-solid/(highly viscous) liquid processable in form of laminable films. The latter, herein deposited on glass substrates, can be used as primary encapsulants for PSCs *via* an industrially compatible solvent- and strain-free lamination protocols, aiming at solving limitations of current approaches based on solid

encapsulants. In addition, we show that the adhesion, barrier and thermal management properties of our homopolymer PIB encapsulant can be improved by the addition of two-dimensional (2D) inorganic fillers, namely few-layer hexagonal boron nitride (*h*-BN) (nano)flakes produced at industrial scale through a patented wet-jet milling exfoliation process of the native bulk powder.^{63,64,65}

The last point is about the figures. The authors are encouraged to improve the figures to make the comparison clearer and easier to read.

We have improved the graphics of all the figures in the revised manuscript to improve their readability.

On the basis of the above consideration, the current form of the manuscript is not suitable for Nature Communications. But if the authors can thoroughly address all the above points, it may be possible to reconsider it.

We thank again the Reviewer for his/her valuable comments on our work. We did our best to reply to all the Reviewer's comments. We believe that our additional characterizations have strengthened the value of our work, which can be now fully appreciated by the Reviewers.

Reviewer #3

*The authors introduce an innovative encapsulation process for perovskite solar cells (PSCs) and perovskite solar modules (PSMs). This process involves the lamination of a highly viscoelastic semi-solid/viscous liquid encapsulant adhesive on the PSC/PSMs. Two types of encapsulants were prepared: a transparent PIB and an opaque composite of PIB with two-dimensional hexagonal boron nitride (*h*-BN) flakes. The encapsulant's viscoelasticity inherently reduces thermomechanical stresses. Additionally, the inclusion of thermally conductive 2D *h*-BN flakes enhances the encapsulant's barrier properties against moisture corrosion. It also improved its thermal management properties. This is good result and can be published in Nature Communications.*

We thank the Reviewer for his/her positive evaluation of our work.

I would suggest minor revision due to the comments below:

1. For Figure 4a left bottom corner, is the '-20 °C' right statement?

We thank the Reviewer for spotting this error. The right temperature is -40°C. **Fig. 4a** has been corrected in the revised manuscript, and is shown here below:

Fig. 4. a) Temperature profile of the thermal shock test performed on the mesoscopic n-i-p PSMs encapsulated with PIB and PIB:h-BN. b) Temperature profile and environmental exposure conditions of the humidity freeze test performed on the mesoscopic n-i-p PSMs encapsulated with PIB and PIB:h-BN. c) PV parameters of the mesoscopic n-i-p PSMs encapsulated with PIB and PIB:h-BN acquired over >200 cycles of the thermal shock test. d) PV parameters of the mesoscopic n-i-p PSMs encapsulated with PIB and PIB:h-BN acquired over >10 cycles of the customized humidity freeze test.

2. For the supporting tables S2-S6, why each condition has the statement 'after 240 h ISOS-D-1 test'?

To ensure data reproducibility, a preliminary ISOS-D-1 test was used as a type of preconditioning for all the PSCs and PSMs investigated in this work. Now that our encapsulants have been validated with various accelerated aging tests, future devices can be directly tested through ageing tests more significant for commercial prospects (including ISOS-D-2, ISOS-L1, thermal cycling and humidity freeze tests).

3. For the module encapsulation part, since the encapsulant (PIB) will transform to a highly viscous liquid, will it get inside the scribing lines? If the answer is yes, how would this penetration affect the module stability?

We thank the Reviewer for this question. Thanks to the semi-solid/liquid nature of our PIB, it certainly manages to penetrate the scribing lines, leading to an optimal lamination process without the presence of air bubbles caused by insufficient coverage of the encapsulants near surface discontinuities. Recently, this advantageous feature is used for the homogeneous and strain-free encapsulation of large perovskite solar panels composed of multiple solar modules, for which other commercially available encapsulants have failed by causing breakdown of the layered cell structure or showing gaps of air or performance degradation after the lamination process. Despite the preliminary nature of these results, we can anticipate that these panels have been already shipped to Crete to extend the current perovskite solar farm, realized by our groups and other collaborators within the EU's funded Graphene Flagship initiative activities on PVs.

4. The encapsulant preparation part mentioned about how to prepare the PIB with and without the *h*-BN. However, it doesn't talk about the volume ratio of the semi-solid PIB (MW 95000) and the solid PIB (MW 800000). Could the author further explain that and why is that ratio?

The encapsulants proposed in our work are based on room temperature highly viscous (Brookfield viscosity >100000 mPas at 10 rpm at temperature < 120 °C) PIB (LMW-80, average molecular weight 95,000) provided by TER Chemicals. In the Methods section, we also mentioned the use of solid PIB (Oppanol N80, average molecular weight 800,000), purchased from BASF. This was initially screened, but then abandoned since it was not properly laminated onto PSCs at target temperature (*e.g.*, <100°C). Also, solid PIB was used as suitable material to evaluate the effect of 2D *h*-BN flakes on the barrier and mechanical properties of the polymeric matrix. The barrier properties of solid PIB and PIB:*h*-BN are shown in **Fig. S1**. The adhesion properties were evaluated through pull-off measurements. Unfortunately, these tests could not carry out on the semi-solid/liquid PIB proposed in this work because of its intrinsic viscoelastic nature. Even though pull-off tests are not reported in the manuscript since referring to other type of PIB (*i.e.*, solid PIB instead of semi-solid PIB), we report here below the outcome for the sake of completeness to this Reviewer.

The pull-off measurements of solid PIB and solid PIB:*h*-BN were performed following the ASTM D4541-02, revealing that the incorporation of 2D *h*-BN flakes into PIB also increases the adhesive strength of the solid PIB homopolymer by 25% (**Fig. R3**), which is consistent with previous studies on polymer: *h*-BN composites (*J. Compos. Mater.*, 2018, 52, 1557–1565).

Fig. R3. Tensile stress curves measured for PIB and PIB:h-BN encapsulants deposited on steel substrates.

5. On page 5, the author mentioned that the PIB has several additives to make it crosslink during the heating process. It means that the PIB will have bonding with the device surface. Will this kind of chemical bonding be weakened or improved after thermal cycling? And will it have volume shrinkage or elongation after heating and cooling?

We thank the Reviewer for his/her important question since our discussion was probably not clear enough. In fact, in our work, we used homopolymer PIB, which is different from the “black” PIB commonly used edge sealer in PV devices, but also as primary encapsulant in PSCs (see ref. *Science*, **2020**, 368, 6497).

We have therefore revised the text to highlight our approach, which, by relying on low-molecular weight viscoelastic (semi-solid)/highly viscous (liquid) homopolymer PIB, avoids cross-linking effects and well as volume shrinkage or elongation after heating and cooling.

The main text now reads:

“In this work, differently from commercially available PIB-based encapsulants commonly used in literature for PSCs,^{40,61,62} we propose low-molecular weight homopolymer PIB as a transparent (viscoelastic) semi-solid/(highly viscous) liquid processable in form of laminable films. The latter, herein deposited on glass substrates, can be used as primary encapsulants for PSCs *via* an industrially compatible solvent- and strain-free lamination protocols, aiming at solving limitations of current approaches based on solid encapsulants. In addition, we show that the adhesion, barrier and thermal management properties of our homopolymer PIB encapsulant can be improved by the addition of two-dimensional (2D) inorganic fillers, namely few-layer hexagonal boron nitride (h-BN) (nano)flakes produced at industrial scale through a patented wet-jet milling exfoliation process of the native bulk powder.^{63,64,65”}

6. Did the author consider and test the oxygen permeability through the PIB?

Unfortunately, oxygen permeation testing analysers are not currently available in our facilities. We tried to overcome this lack by specific electrochemical tests, hardly reported in PV

applications but giving important information regarding the barrier properties of coatings. Prospectively, we are working to extend our characterizations to both oxygen transmission rate and helium leak analyses, relying on our involvement in projects on PSCs including partners with such equipment. Additionally, the effect of specific additives, such as tackifiers, on the proposed PIB and PIB:*h*-BN encapsulants will be subject matter of future studies to refine our encapsulation towards the establishment of market-ready products.

REVIEWER COMMENTS

Reviewer #1 (Remarks to the Author):

The authors have resolved most of my confusion, but there are still some issues in the manuscript. Therefore, I suggest that the authors further revise the issues in the manuscript. The problems are as follows :

1. The authors evaluated the impact of 2D h-BN flakes on thermal management performance through infrared thermal imaging, but only the temperature over time curve (100s-200s) was found. The reviewer suggests that the authors supplement the complete temperature change curve throughout the entire process and add representative infrared photos.
2. The testing time of the devices in Fig. S4 and Fig. S6 figure are all >1000 h, while the annotation states that they are >2000 h. Please carefully check.
3. For the mesoscopic n-i-p PSCs based on spiro-OMeTAD HTLs, the encapsulated cells have shown T80 >1000 h during the ISOS-D-2 test in Fig. S4. Due to the poor thermal stability of spiro OMeTAD, the reviewer believes that the stability is difficult to achieve. Please explain the reason.
4. The reviewer suggests supplementing the data on adhesion performance in the Supporting Information.
5. Will the introduction of BN affect the performance of PIB in suppressing lead leakage? The reviewer suggests that the authors supplement lead leakage data for PIB-encapsulated devices in Fig. S9.

Reviewer #2 (Remarks to the Author):

The revised version has addressed all my concerns in my original report. I support publication of this paper as is.

Reviewer #3 (Remarks to the Author):

my comments have been well addressed by the authors. I donot have other concerns.

Below we report the point-to-point answer to the Reviewers' comments.

Reviewer #1

The authors have resolved most of my confusion, but there are still some issues in the manuscript. Therefore, I suggest that the authors further revise the issues in the manuscript. The problems are as follows :

We thank the Reviewer for his/her evaluation of the revision of our manuscript. We carefully considered his/her comments, which helped us to improve the quality of our work. We do hope that the Reviewer can fully appreciate our study.

- 1. The authors evaluated the impact of 2D h-BN flakes on thermal management performance through infrared thermal imaging, but only the temperature over time curve (100s-200s) was found. The reviewer suggests that the authors supplement the complete temperature change curve throughout the entire process and add representative infrared photos.*

We have followed the valuable suggestion of the Reviewers, complementing temperature data over the entire duration of the test and reporting representative infrared thermal images acquired after 60 s of cooling.

The revised **Fig. S3** is reported here below:

Fig. S3. a) Maximum temperature over time measured for the glass/PIB/glass and glass/PIB:*h*-BN/glass systems (area = 5.6 cm×5.6 cm) first heated at 90°C (t = 0 s) and then transferred to an Al platform at 25°C. The internal panel shows the magnification of the figure in the 100-200 s time interval. b) Infrared thermal images of glass/PIB/glass and glass/PIB:*h*-BN/glass systems acquired after 60 s of cooling.

2. *The testing time of the devices in Fig. S4 and Fig. S6 figure are all >1000 h, while the annotation states that they are >2000 h. Please carefully check.*

We thank the Reviewer for spotting these inaccuracies. The captions of **Fig. S4** and **Fig. S6** have been amended in the revised manuscript.

The captions now read:

“Fig. S4. a) Sketch of the structure of the large-area (1 cm²) mesoscopic n-i-p PSCs based on Cs_{0.08}FA_{0.80}MA_{0.12}Pb(I_{0.88}Br_{0.12})₃ perovskites and spiro-OMeTAD HTLs. b) JV curves measured for the as-fabricated mesoscopic n-i-p PSCs based on spiro-OMeTAD HTLs before and after encapsulation with PIB:*h*-BN (before and after 240 h-ISOS-D-1). c) PV parameters of the investigated mesoscopic n-i-p PSCs based on spiro-OMeTAD HTLs acquired over >1000 h of ISOS-D-2 test.

Fig. S6. a) Sketch of the structure of the large-area (1 cm²) inverted p-i-n PSCs based on PTAA HTLs and PCBM ETLs. b) JV curves measured for the as-fabricated inverted p-i-n PSCs before and after encapsulation with PIB:*h*-BN (before and after 240 h-ISOS-D-1). c) PV parameters of the investigated inverted p-i-n PSCs acquired over >1000 h of ISOS-D-2 test.”

3. *For the mesoscopic n-i-p PSCs based on spiro-OMeTAD HTLs, the encapsulated cells have shown T80 >1000 h during the ISOS-D-2 test in Fig. S4. Due to the poor thermal stability of spiro OMeTAD, the reviewer believes that the stability is difficult to achieve. Please explain the reason.*

We have appreciated this Reviewer’s comment. It has been demonstrated that LiTFSI, used as spiro-OMeTAD dopant, is hygroscopic and has the tendency to absorb moisture from air and aggregate it into spiro-OMeTAD, causing phase segregation (*Energy Environ. Sci.*, **14**, 5161–5190 (2021); *Adv. Energy Sustain. Res.*, **3**, 2200045 (2022)). This effect causes changes in the spiro-OMeTAD energy levels and morphology, including the formation of pinholes. In addition, tBP, used as spiro-OMeTAD dopant, has a low glass transition temperature and evaporates at high temperatures, leading to morphological changes, including pinhole formation, in spiro-OMeTAD HTLs (*Energy Environ. Sci.*, **14**, 5161–5190 (2021); *Adv. Energy Sustain. Res.*, **3**, 2200045 (2022)). In this scenario, the formation of pinholes in spiro-OMeTAD creates pathways through which iodide ions penetrate the spiro-OMeTAD layer, reacting with metallic electrodes as well as reducing the doped spiro-OMeTAD, which, in turn, decreases its conductivity (*Energy Environ. Sci.*, **14**, 5161–5190 (2021); *Adv. Energy Sustain. Res.*, **3**, 2200045 (2022)). Also, pinholes can be deep enough to establish a direct connection between the CTLs, forming shunting pathways. In this context, the use of ultrathin perovskite-passivating layers, like PEAI in our case, increases the crystallinity of the perovskite at the interface with the spiro-OMeTAD, inhibiting morphological changes. More in detail, PEAI effectively passivate defects and trap states at the

perovskite/spiro-OMeTAD interface, helping our cell' structure to endure the lamination process at 90°C for 10 min while using spiro-OMeTAD as HTL (*Nat. Photonics*, **13**, 460–466 (2019)). However, our unencapsulated cells still exhibited significant PCE loss (>20%) after 150 h during ISOS-D2. This highlights the critical role of the air moisture in doped Spiro-OMeTAD degradation through oxidation (*ACS Appl. Energy Mater.*, **4**, 12, 13696–13705 (2021)).

Encouragingly, encapsulated devices based on spiro-OMeTAD HTLs lost 20% of their initial PCE around 1000 h. This excellent stability is attributed to the sturdiness of our encapsulants, which effectively protects against the air/moisture ingress into the cell structure. Our findings are well aligned with existing research (*Adv. Mater.*, **36**, 2308039 (2024); *Adv. Funct. Mater.*, **31**, 2100557 (2021); *Nat. Commun.*, **13**, 7639 (2022)), and strongly suggest that our encapsulation strategy paves the way for improving the stability of efficient PSCs based on spiro-OMeTAD HTLs. More in detail, *Adv. Mater.*, **36**, 2308039 (2024) shows an ISOS-D3 test for unencapsulated spiro-OMeAD-based device that reached a T₉₀ of 1200 h; *Adv. Funct. Mater.*, **31**, 2100557 (2021) shows an impressive stability (T₈₀ of 3600 h) for spiro-OMeTAD-based PSC. Table 16 of the Supporting Information of *Nat. Commun.*, **13**, 7639 (2022) reports an unencapsulated mesoporous PSC based on spiro-OMeTAD HTL with a T₈₀ of 1096 h when kept at 80°C. Overall, we think that the combination of advanced encapsulants with interface engineering is a promising strategy to stabilize high temperature-operating PSCs based on spiro-OMeTAD HTLs. Once again, thank you for bringing attention to these critical considerations in stabilizing spiro-OMeTAD-based PSCs, and for providing valuable insights into potential solutions. Your input contributes to the ongoing efforts to concomitantly improve the stability and performance of PSCs.

The main text has been revised and now reads:

“In general, the hygroscopicity of lithium bis(trifluoromethanesulfonyl) imide (LiTFSI) and the evaporation of 4-tert-butylpyridine (tBP), used as spiro-OMeTAD dopants, promote moisture entry into the cell structure and morphological changes of both perovskite and spiro-OMeTAD HTL.⁸¹ These effects are accelerated with increasing temperature, leading to the formation of pinholes that accelerate iodine migration to iodine-sensitive cellular components (*e.g.*, metal electrodes) and even cause connections between CTLs (shunting pathways), leading to PCE losses.⁸¹ In this context, our results demonstrate that the combination of advanced PIB-based encapsulants, which block moisture entry into PSCs, and ultrathin perovskite passivation layers, *e.g.*, PEAI, is a promising strategy to stabilize spiro-OMeTAD-based PSCs operated at high temperature. In particular, PEAI effectively passivate defects and trap states at the perovskite/spiro-OMeTAD interface,⁸⁹ helping our spiro-OMeTAD-based cell to withstand the lamination process at 90°C for 10 min. However, our data indicate that proper encapsulants that effectively protects against the air/moisture ingress into the cell structure are crucial to avoid doped spiro-OMeTAD degradation through oxidation.⁹⁰ Our findings are well aligned with existing studies reporting excellent thermal stability of spiro-OMeTAD-based PSCs.^{91,92,93”}

4. *The reviewer suggests supplementing the data on adhesion performance in the Supporting Information.*

The adhesion properties were evaluated through pull-off measurements. Unfortunately, these tests could not be carried out on the semi-solid/liquid PIB proposed in this work because of its intrinsic viscoelastic nature. Pull-off tests were therefore measured on solid PIB and *h*-BN films.

The main text now reads:

“The adhesive properties of solid PIB and PIB:*h*-BN films were measured through pull-off tests following the ASTM D4541-02, showing that the incorporation of 2D *h*-BN flakes into PIB increases the adhesive strength of the homopolymer PIB by 25%.⁶⁵ Notably, both water contact angle and pull-off measurements were carried out on solid PIB and PIB:*h*-BN films since semi-solid/liquid films do not permit reliable measurements with these techniques.”

“The adhesive properties of solid PIB and solid PIB:*h*-BN were measured through pull-off tests using an Instron 3365 dual-column dynamometer equipped with a 2 kN load cell and following the ASTM D4541-02 standard. The encapsulant resins were deposited on steel plates, which were clamped to the bottom anvil. Afterwards, the 15 mm diameter top piston was painted with cyanoacrylate adhesive and immediately put in contact with the sample. A force of 15 N was applied to the sample and the adhesive was let curing for 30 min. Normal displacement was then applied to the piston, with a rate of 1 mm min⁻¹ until separation.”

Fig. S2 was revised and now includes pull-off measurements, as shown here below:

Fig. S2. Photographs of a water drop on the surface of the a) solid (high-molecular weight) PIB and b) PIB:h-BN films. c) Water contact angle data measured for the solid PIB and PIB:h-BN films. d) Tensile stress curves measured for solid PIB and PIB:h-BN films deposited on steel substrates. The films were produced with solid (high-molecular weight) PIB to avoid gravity-induced flatness alteration in viscoelastic films, which may lead to unreliable water contact angle results. Also, solid films were needed for reliable pull-off measurements. Water contact angle data were reproduced from ref. 65 of the main text (Molina-Garcia, M. A. et al., *J. Phys. Mater.* **6**, 035006 (2023)). Solid PIB (Oppanol N80, average molecular weight 800,000) was purchased from BASF.

5. Will the introduction of BN affect the performance of PIB in suppressing lead leakage? The reviewer suggests that the authors supplement lead leakage data for PIB-encapsulated devices in Fig. S9.

The Pb leakage-inhibiting ability measured for PIB encapsulants was comparable to that measured for PIB:*h*-BN. Additional ICP-OES measurements were included in the revised manuscript which now reads:

“The effectiveness of the PIB:*h*-BN encapsulant to protect the PSMs from extrinsic factors was also assessed by measuring the Pb leakage of the encapsulated PSM immersed in water through inductively coupled plasma optical emission spectroscopy (ICP-OES) (Fig. S9). After water immersion the unencapsulated PSM rapidly degraded, showing yellowing associated with the decomposition of the perovskite to PbI₂. Because of its high solubility (340 mg L⁻¹, solubility product constant = 4.4×10⁻⁹ M)^{57,58} PbI₂ rapidly dissolved in water, causing cracking of the Au rear electrode. The detected Pb leakage (>60 μg cm⁻² after 24 h) is consistent with the Pb content in perovskite, typically between 0.1 and 1 g m⁻².⁵⁶ Contrary to unencapsulated devices, the perovskite in the encapsulated PSMs retained its starting colour, preserving the perovskite phase. Consequently, the Pb leakage was drastically inhibited to values lower than 1 μg cm⁻² after 24 h (low Pb water contamination is likely associated with perovskite residuals nearby the encapsulant edges and not with the degradation of perovskite over the PSM active area). Similar Pb leakage inhibition was observed for a PIB encapsulant protecting simple perovskite films.”

Fig. S9 was revised to include supplementary ICP-OES data obtained with PIB encapsulants, as shown here below:

Fig. S9. a,b) Photograph of the unencapsulated PSM (rear side) and a PSM encapsulated with PIB:*h*-BN (rear and front side) after 24 h of immersion in water. c) Areal Pb leakage from the

investigated PSMs over water immersion time. The Pb leakage from a perovskite film encapsulated with PIB was also measured (sample named PIB).

Reviewer #2

The revised version has addressed all my concerns in my original report. I support publication of this paper as is.

We thank again the Reviewer for taking the time to review our manuscript.

Reviewer #3

my comments have been well addressed by the authors. I do not have other concerns.

We thank the Reviewer for his/her positive evaluation of our revised work.

REVIEWERS' COMMENTS

Reviewer #1 (Remarks to the Author):

The author answered the questions well now.